# LOX-1 in Cardiovascular Disease: A Comprehensive Molecular and Clinical Review

**DOI:** 10.3390/ijms25105276

**Published:** 2024-05-12

**Authors:** Maria Eugenia Sánchez-León, Karen Julissa Loaeza-Reyes, Carlos Alberto Matias-Cervantes, Gabriel Mayoral-Andrade, Eduardo L. Pérez-Campos, Laura Pérez-Campos-Mayoral, María Teresa Hernández-Huerta, Edgar Zenteno, Yobana Pérez-Cervera, Socorro Pina-Canseco

**Affiliations:** 1Centro de Investigación Facultad de Medicina-UNAM-UABJO, Universidad Autónoma “Benito Juárez” de Oaxaca, Oaxaca 68020, Mexico; maeugmels.208305@gmail.com (M.E.S.-L.); karen.loaeza.r@gmail.com (K.J.L.-R.); carloscervantes.ox@outlook.com (C.A.M.-C.); drmayoral@gmail.com (G.M.-A.); laupcm9@gmail.com (L.P.-C.-M.); 2Centro de Estudios en Ciencias de la Salud y la Enfermedad, Facultad de Odontología, Universidad Autónoma “Benito Juárez” de Oaxaca, Oaxaca 68020, Mexico; 3Tecnológico Nacional de México/IT Oaxaca, Oaxaca 68030, Mexico; perezcampos123@yahoo.es; 4Consejo Nacional de Humanidades, Ciencias y Tecnologías, Facultad de Medicina y Cirugía, Universidad Autónoma “Benito Juárez” de Oaxaca, Oaxaca 68120, Mexico; marte-hh28@hotmail.com; 5Facultad de Medicina, Universidad Nacional Autónoma de México, Mexico City 04510, Mexico; ezenteno@unam.mx

**Keywords:** LOX-1, ox-LDL, cardiovascular diseases, atherosclerosis, endothelial dysfunction, metabolic disorders

## Abstract

LOX-1, ORL-1, or lectin-like oxidized low-density lipoprotein receptor 1 is a transmembrane glycoprotein that binds and internalizes ox-LDL in foam cells. LOX-1 is the main receptor for oxidized low-density lipoproteins (ox-LDL). The LDL comes from food intake and circulates through the bloodstream. LOX-1 belongs to scavenger receptors (SR), which are associated with various cardiovascular diseases. The most important and severe of these is the formation of atherosclerotic plaques in the intimal layer of the endothelium. These plaques can evolve into complicated thrombi with the participation of fibroblasts, activated platelets, apoptotic muscle cells, and macrophages transformed into foam cells. This process causes changes in vascular endothelial homeostasis, leading to partial or total obstruction in the lumen of blood vessels. This obstruction can result in oxygen deprivation to the heart. Recently, LOX-1 has been involved in other pathologies, such as obesity and diabetes mellitus. However, the development of atherosclerosis has been the most relevant due to its relationship with cerebrovascular accidents and heart attacks. In this review, we will summarize findings related to the physiologic and pathophysiological processes of LOX-1 to support the detection, diagnosis, and prevention of those diseases.

## 1. Introduction

LOX-1, also known as ORL-1 or lectin-like oxidized low-density lipoprotein receptor one, is the primary receptor for oxidized low-density lipoproteins (ox-LDL) circulating in the bloodstream, originating from food intake. It belongs to the scavenger receptors (SR) family [1], which are associated with various cardiovascular pathologies. The most important and severe of these is the formation of atherosclerotic plaques in the intimal layer of the endothelium, which can evolve into complicated thrombi with the participation of fibroblasts, activated platelets, apoptotic muscle cells, and macrophages transformed into foam cells. This process causes changes in vascular endothelial homeostasis, leading to partial or total obstruction in the lumen of blood vessels, resulting in reduced blood flow to the heart muscle and myocardial ischemia [2]. LOX-1 is responsible for the formation of foam cells (with the participation of differentiated monocytes), a mediator between the internalization and degeneration of endothelial cells, as well as the initiator of myocardial ischemia, which does not show symptoms in its development until the deposition is formed and oxygen deprives to the heart [3]. The main objective of this review is to provide crucial information about LOX-1 to support the detection, diagnosis, and prevention of mentioned pathologies.

## 2. Molecular Characteristics of LOX-1

The lectin-like low-density lipoprotein receptor LOX-1 is a 50 kDa transmembrane glycoprotein (52 kDa in *Homo sapiens*) [4]. Initially, it was identified in bovine endothelial cells from the aorta, where it is the receptor for the binding, internalization, and degradation of ox-LDL [5]. LOX-1 belongs to the C-type lectin superfamily and comprises 273 amino acids [6]. It has cytoplasmic, transmembrane, and extracellular domains [7] and is expressed in macrophages, vascular smooth muscle cells, cardiomyocytes, platelets, and fibroblasts [8,9]. LOX-1 has a secondary role by contributing to endothelial dysfunction and apoptosis. Consequently, the seclusion of ox-LDL forms foam cells in macrophages and vascular smooth muscle cells and platelets. LOX-1 is encoded by the OLR gene in human cells, is inducible, and has homology to the NK gene in natural killer cells. The OLR gene is 7000 base pairs long and comprises 6 exons and 5 introns, showing a single copy in the chromosome region p12.3–13.2, containing the TATA (29 base pairs) and CAAT (99 base pairs) boxes on the 5′ flanking side [6].

The hypothetical structure of LOX-1 is described in Figure 1.

After synthesis translation, LOX-1 undergoes post-translational modifications as N-glycosylation, which is the glycosidic linkage of GlcNAc to NH2 from asparagine. The *N*-glycosylation process includes two principal phases: assembling a lipid-linked oligosaccharide (LLO) and transferring the oligosaccharide to selected asparagine residues of polypeptide chains [10]. This glycosylation seems to be essential in eukaryotic organisms since various functions are related to this process, such as ligands in various recognition processes to stabilize proteins in denaturation and proteolysis, modulate the immune response, participate in the relative orientation of proteins to a membrane, and give the structural rigidity properties in their union with other proteins, regulate the protein turnover (intracellular traffic and provide stability and folding of proteins) and mediate interactions with pathogens transferred [11,12]. 

To find out if the binding of LOX-1 with its receptor is dependent on the glycosylation of LOX-1 and therefore may have an essential role in atherogenesis, a study was carried out, performing pulse-chase labeling and digestion with glycosidases, finding an immature form of the receptor, called pre-LOX-1 of 40 kDa with glycosylated chains (high proportion of mannose), which, as it matures, acquires more carbohydrates, weighting 48 kDa. Likewise, the inhibition of glycosylation with tunicamycin is also carried out in cells activated with TNF-α, obtaining a deglycosylated gene. It was demonstrated using a cellular enzyme-linked immunosorbent assay, flow cytometry, and immunofluorescence confocal microscopy that the deglycosylated form of LOX-1 is stored in the endoplasmic reticulum and/or Golgi apparatus in bovine aortic endothelial cells; otherwise, the opposite occurred in BLOX-1-CHO cells, which was expressed on the cell surface but with a low affinity towards its ligand [13].

In another study, the relationship between the structure and function of LOX-1 was analyzed when mutagenesis studies were carried out. It was demonstrated that the carbohydrate recognition domain is the site where the binding with ox-LDL takes place, so we also performed the analysis of this recognition domain, finding that the six cysteines in the intrachain disulfide bonds are essential for shaping the folding and actively participating in processing and transportation. For this protein, the terminal sequence KANLRAQ is necessary for the folding and transport of proteins and the four terminal residues are involved in the maintenance of the receptor function [14]. 

### 2.1. Polymorphisms, Mutations, and Isotypes of LOX-1

Mutations and polymorphisms describe DNA variants, double helixes, or epigenetic replication errors caused by external environmental factors like prolonged exposure to UV rays, certain chemicals, or pathogens. Mutation refers to any change in the base sequence of DNA, i.e., a heritable change in the genome [15]. In contrast, polymorphism refers to DNA variants within a population; the main difference is that a mutation is a change in the DNA sequence of an organism’s genome and a polymorphism is a mutation that occurs in more than 1% of a specific population and can be hereditary or acquired [16]. Various population factors promote the development of pathologies where the expression of LOX-1 is involved, as explained below.

The Turkish population concludes that high plasma levels of sLOX-1, ox-LDL, LOX-1 K167N, and 3′UTR188CT are significantly associated with the risk of developing preeclampsia, proposing sLOX-1 as a therapeutic target for this condition in pregnant women. [17]. These two polymorphisms have also been analyzed by Birsen Aydemir et al. in Turkish pregnant patients with gestational diabetes mellitus. The results revealed that these genotypic presentations seem to be related to the increase in oxidative stress in pregnant patients with diabetes mellitus [18]. In several studies reviewed, polymorphisms are typical of specific populations and interests, for which research has been carried out in subjects with specific characteristics, for example, the rs1050283 polymorphism of the 3′ LOX-1 untranslated region studied in a Chinese population of 526 patients with cerebral infarction due to atherosclerosis were compared to 640 healthy subjects, using DNA sequencing, real-time PCR, and Western blotting to know the expression of LOX-1 as well as the ELISA technique to quantify sLOX-1, finding that polymorphism of the allele of LOX-1 shows a strong association with the increase in the risk of brain infarction by atherosclerosis and affects the levels of LOX-1 and sLOX-1 [19].

There is a beneficial variant, the immature form of LOX-1, named LOXIN, an isoform that lacks a portion of the functional domain of LOX-1. This variant is produced by single nucleotide polymorphism (linkage disequilibrium block of SNPs), in which a variation in a single base pair occurs in the intronic sequence of the OLR1 gene, where the alternative splicing of the OLR1 mRNA is modulated, giving rise to different proportions of the complete LOX-1 and LOXIN receptor. It has been proposed that the LOXIN isoform exerts a protective function by blocking the LOX-1 activation [20]. LOX-1 is encoded by OLR1 (oxidized low-density lipoprotein lectin-like receptor 1 gene located on human chromosome 12 in the p12.3–p13.2.8 region) [7]. Studies carried out in vitro in cultured bovine aortic cells and isolated female Chinese hamster cells showed that the approximate molecular weight is 35 kDa derived from LOX-1 expressed on the cell surface. The N-terminal amino acid sequencing of the soluble form indicated two cleavage sites in Arg86-Ser87 and Lys89-Ser90, located in the extracellular domain near the membrane. The sLOX-1 is a therapeutic target for atherosclerotic diseases [21] and acute cerebral infarction [22,23]. It is associated with cardiovascular and cerebrovascular events in patients undergoing percutaneous coronary surgery [24,25] and is even related to inflammatory diseases such as sleep apnea [26]; moreover, in patients with rheumatoid arthritis (the soluble portion of LOX-1 has been identified in synovial fluid) [27] and obesity [28]. 

Another alternative isoform of LOX-1 that has been identified is LOX-1Δ4 or LOX-1 splice variant; this isoform has the function of increasing the proliferation rate of the non-tumorigenic epithelial cell line, MCF12-F, that is, it acts as an oncogene and negatively regulates cell death proteins (conversion of non-tumorigenic cells to tumorigenic cells) and strongly modulates histone H4 and Ku70 acetylation (regulation and maintenance of chromatin structure in the cell as well as DNA repair), which reduces the rate of cell death. (a limiting factor of the DNA double-strand break repair machinery), implicated in the inhibition of apoptosis and resistance to drugs. In addition, LOX-1 is overexpressed in 70% of human cancers and is positively correlated with tumor stage and grade, suggesting LOX-1 is a therapeutic target for breast cancer treatment [29]. LOX-1Δ4 was reported in 2012 as lacking the W150A region; this region is present in the C lectin-like domain or CTLD; this domain is a canonical dimer with essential spine characteristics on its surface. The canonical CTLD dimer is formed through tight hydrophobic interactions in the intact LOX-1 receptor, where a modified or mutated portion of the CTDL domain, called W150A, participates in the primary interaction with the ox-LDL binding site. The loss of tryptophan in the ring due to a mutation to alanine prevents the formation of the canonical dimer [29].

Another amino acid dimorphism (Lys/Asp) at residue 167 of the LOX-1 receptor studied in Turkish subjects is produced by changing a single nucleotide K167N (G501C) of the OLR-1 gene. This Lys replacement decreases the binding and internalization of ox-LDL, concluding that this polymorphism seems independent of other cardiovascular risk factors, such as smoking and male gender, among others [30]. 

How do specific mutations or isoforms of LOX-1 contribute to disease pathogenesis or progression?

The LOX-1 receptor is crucial in developing atherosclerosis and related cardiovascular conditions. Mutations and alternative splicing resulting in different isoforms of LOX-1 can significantly influence the receptor’s function, thereby affecting disease pathogenesis and progression.

#### 2.1.1. Mutations of LOX-1

Impact on LOX-1 Functionality: Mutations in the OLR1 gene can alter the receptor’s structure and function. For example, specific mutations may affect the receptor’s ability to bind to ox-LDL, either enhancing or inhibiting this critical interaction. Enhanced binding capacity can lead to increased formation of foam cells, accelerating atherosclerotic plaque development. Conversely, mutations that reduce LOX-1’s affinity for ox-LDL could confer protective effects against atherosclerosis.Clinical Relevance of Mutations: Specific mutations in the OLR1 gene have been associated with an increased risk of cardiovascular diseases. For instance, specific polymorphisms linked to the OLR-1 gene have been correlated with higher susceptibility to coronary artery disease and myocardial infarction. Understanding these genetic variations can help identify individuals at higher risk and lead to targeted prevention strategies.

#### 2.1.2. Isoforms of LOX-1

LOXIN Isoform: An important alternative splicing event within the OLR1 gene leads to the production of an isoform known as LOXIN. This isoform lacks a portion of the C-terminal domain, crucial for the receptor’s ability to bind ox-LDL. LOXIN acts as a dominant-negative regulator of the full-length LOX-1 receptor, potentially reducing ox-LDL uptake and subsequent foam cell formation. This suggests that higher levels of LOXIN relative to full-length LOX-1 could be protective against atherosclerosis;Therapeutic Implications of LOXIN: Enhancing the expression of LOXIN or administering it as a therapeutic protein could provide a novel approach to preventing or treating atherosclerosis. By competing with the full-length LOX-1 for binding sites, LOXIN could reduce ox-LDL uptake and the ensuing inflammatory responses in the vascular endothelium.

In conclusion, the research on mutations and isoforms of LOX-1 reveals their importance in cardiovascular diseases. These variations offer a deeper understanding of these diseases’ pathogenesis and suggest new therapeutic opportunities. Strategies that modulate the expression levels of LOXIN or mimic its effects could present innovative approaches for preventing and treating atherosclerosis and related conditions. Studying these mutations and isoforms provides a solid foundation for more personalized and effective therapeutic interventions in cardiovascular diseases, opening up new avenues for improving overall cardiovascular health.

In the following paragraph, we highlight the importance of LOX-1 at a physiological level.

## 3. Physiological Functions of LOX-1

Macrophages regulate vascular homeostasis in blood circulation cells and express scavenger receptors; therefore, it is essential to highlight that LOX-1 is the central scavenger receptor of ox-LDL but not the only one. The participation of SR-A1 (scavenger Receptor A1), CD36 (Platelet Glycoprotein—scavenger receptor type B), and LOX-1 has been reported, and some other molecules take part as intermediaries in the seclusion of cholesterol in the cell, such as ABCA1, ABCG1, and SR-BI in the reverse metabolism of cholesterol [31]. The enzyme acetyl-coenzyme A cholesterol transferase (ACAT1) has the function of esterifying cholesterol, resulting in cholesterol esters that are transported as droplets to the rough endoplasmic reticulum, acting on the hydroxy-methyl-Glutaryl CoA reductase, also called the rate-limiting enzyme, which acts on the synthesis of the LDL receptor to maintain the energy requirements in the cell as presented in Figure 2. However, in pro-atherosclerotic and proinflammatory stages, the expression of LOX-1 increases, increasing ACAT1 [32] and producing seclusion of free cholesterol and cholesteryl esters in macrophages, giving rise to foam cells, as well as vascular endothelial and muscle cells that can transform into foam cells, producing lipid encrustations in the intimal wall, obstructing the lumen of blood vessels [31].

An analysis realized that the LOX-1 receptor was the cloning of the cDNA, which mediates the action of ox-LDL in the endothelial cells of the capillary vessels. This study was carried out using Dil-OxLDL (oxidized human plasma low-density lipoprotein, labeled with Dil-1,1′-dioctadecyl-3,3,3′,3′-tetramethylindocarbocyanine perchlorate-) via the cell sorting technique (FACS), finding that endothelial cells in vivo and in vitro internalize and degrade ox-LDL through a receptor that does not involve macrophage scavengers [5].

## 4. Pathological Function of LOX-1

Endothelial cells, macrophages, smooth muscle cells, platelets, and cardiomyocytes have been reported to express LOX-1 [33]. At the origin of the atherosclerotic plaque, an accumulation of collagen coexists. Likewise, macrophages, smooth muscle cells, and cells of the vascular intimate layer (endothelium) are involved; this condition is of pathophysiological origin [34]. In another work, the possible link between LOX-1, mitochondrial DNA damage, autophagy, and NLRP3 inflammasome expression in human THP-1 macrophages exposed to lipopolysaccharide was studied in vitro, finding that LOX-1 expression was exacerbated. The generation of reactive oxygen species (ROS), autophagy, damage to mitochondrial DNA, and the NLRP3 inflammasome was corroborated by adding a binding antibody to LOX-1, inhibited effectively inhibiting the abovementioned processes [35]. 

In vivo studies have found that the expression of the LOX-1 receptor is upregulated in the presence of pathological conditions such as atherosclerosis, hypertension [36], and diabetes. It is well-studied that in endothelial cells, LOX-1 is overexpressed in the early stages of atherogenesis and the late stages in de novo vascular formations [8]. In other studies, carried out in an in vivo model, the relationship between the expression of LOX-1 and arterial hypertension was studied, where it was shown that LOX-1 is overexpressed in the aorta and vein of hypertensive rats previously fed with a diet high in salt but not in the comparative rats, so it was suggested that LOX-1 had a possible role in the pathogenesis of hypertension in the said animal model [36,37,38,39].

### 4.1. LOX-1 Proinflammatory Inductors 

Various factors stimulate the expression of LOX-1 [8] such as proinflammatory cytokines (tumor necrosis factor TNFα) [33], interleukin-1 (IL-1α and IL-1β), interferon-gamma (IFNγ) [34], lipopolysaccharide (LPS) [34], C-reactive protein (CRP) [40], modified lipoproteins (Ox-LDL) [40,41,42], modified LDL 15-lipoxygenase [43], oxidized HDL Lipoxygenase [44], glycosylated LDL [37], lysophosphatidylcholine [38], palmitic acid [39], stimuli related to hypertension (angiotensin II and stimuli such as shear stress) [45], hyperglycemic stimuli including high glucose, advanced glycation end products (AGEs [46]) and other stimuli (homocysteine [47] and free radicals [48]). Therefore, it is expected that LOX-1-inducing agents will favor the hyperactivation of the endothelium in response to low-grade inflammation, followed by an increase in the permeability of lipoproteins that will be accompanied by an oxidative environment that will favor the production of ox-LDL; with this, the monocytes will transform into macrophages. These will recruit low-density lipoproteins (LDL) to form foam cells, which will adhere to the arterial intimal layer. In more advanced stages, it becomes complicated, generating an endothelial rupture. To repair the damage, the cell sends signals to trigger platelet adhesion, producing a thrombus; the smooth muscle cells migrate to the damaged area and, in turn, the thrombus obstructs the lumen of their capillaries, producing heart attacks and stroke.

#### 4.1.1. Proinflammatory Cytokines and LOX-1

Proinflammatory cytokines are positively regulated by the expression of the toll-like receptor 4 that triggers the activation of nuclear factor kappa B (NF-κB) in different tissues; thus, with the promotion of inflammation, apoptosis, and autophagy are increased. The signaling pathway is turned on when the proprotein convertase subtilisin/kexin type 9 (PCSK9), which is a plasma protein, regulates the expression of the toll-like receptor; it has been studied that it is linked with some inflammatory mediator scavenger receptors, such as LOX-1, exacerbating the inflammatory response, thus showing an inflammatory response in atherosclerosis and, with it, the expression of these cytokines [49]. In the next paragraph, we talk about the proinflammatory mediators that intervene in the expression of LOX-1.

In studies where the regulation of LOX-1 expression in smooth muscle cells was analyzed, the participation of proinflammatory mediators of the acute phase, such as IL-1α, IL-1β, and TNF-α, whose correlation in the expression of LOX-1 is partially related to advanced atherosclerotic lesions, was determined using PCR and Western blot [34]. On the other hand, IL-1B has been found to promote the absorption of oxidized LDL in human glomerular mesangial cells with the prominent participation of LOX-1, suggesting that a therapy to lower LOX-1 levels is a promising therapy to treat glomerulosclerosis that is related to renal failure in young and adult patients [50]. Likewise, the relationship with the pathology of type 2 diabetes mellitus is treated using endothelial progenitor cells, finding that the expression of LOX-1 triggers the activation of the NLRP3 inflammasome as well as the inhibition of LOX-1 by drugs that suppress the activation of the inflammasome and therefore promotes the blockade of the adaptor protein ASC associated with apoptosis, which in turn leads to the cleavage and activation of caspase 1, whose domino effect cleaves il-1b precursors related to pathogenesis and vascular complication in type 2 diabetes mellitus [51]. In another study, two molecules involved in the rho signaling pathway (ARHGEF1 and ROCK2) were identified, finding that ROCK2 is dynamically associated with LOX-1 when ox-LDL is present, stimulating the catalytic activity of ROCK2. At the same time, inhibition of ROCK2 reduces the activation of NF-κB and, consequently, the production of IL-8. This process is a mechanism that potentially contributes to atherogenesis through the disruption of vascular homeostasis by endothelial dysfunction [52]. Some studies have suggested that intracellular signaling events occur depending on the activation of the LOX-1 receptor. The generation of reactive oxygen species inactivates nitric oxide by reducing the endothelial activity of nitric oxidase synthase (Enos). However, in vivo, the result is an imbalance in vascular homeostasis, which leads to the activation of NF-κB [53].

In summary, LOX-1 plays a significant role in the pathogenesis of atherosclerosis and other cardiovascular diseases. LOX-1 facilitates the ox-LDL by endothelial cells and macrophages, forming foam cells and developing atherosclerotic plaque. This process is central to endothelial dysfunction, a key event in the initiation and progression of atherosclerosis. Additionally, LOX-1 is implicated in the vascular inflammation that characterizes these pathologies, making it a potential target for therapeutic intervention.

#### LOX-1 Function in Atherosclerosis and Related Diseases

Excessive ROS production contributes to atherosclerosis and, in the presence of lipid peroxidation and DNA damage, is a factor in cancer development. Activation of LOX-1 stimulates the expression of proinflammatory and proangiogenic molecules, such as NF-kB and VEGF, in endothelial cells and macrophages. LOX-1 is implicated in cardiovascular diseases and various types of cancer. Inhibition of LOX-1 has been shown to reduce NF-κB activation and inflammatory pathways, suggesting a link between atherosclerosis and cell transformation. Additionally, remodeling proteins like MMP-2 and MMP-9 increase angiogenesis in atherosclerotic plaque and human prostate cancer cells. Inhibiting LOX-1 expression could represent a promising strategy for treating atherosclerosis and cancer [54].

In hypertension, it has been demonstrated that hypertensive patients are more susceptible to LDL oxidation, have high levels of Ang II, and have a higher incidence of myocardial infarction. In contrast, normotensive controls have a lower susceptibility to LDL oxidation. A disadvantage in hypertensive patients is that, due to high levels of Ang II, they have a greater direct induction of oxidative stress and inflammatory response in the aorta, thus increasing the uptake of ox-LDL by endothelial cells through positive regulation of LOX-1, leading to endothelial damage, which are responsible for maintaining vascular tone and whose molecules are involved in such events are prostacyclin (PGI2), endothelin (ET), Ang II, and NO. Vasomotor tone regulation appears to be controlled by the constant action of NO. Inhibition of NO and PGI2 formation allows opposing forces of vasodilation, such as ET, Ang II, or thromboxane A2, to determine the artery’s ability to maintain its lumen in the presence of changing forces caused by the formation and progression of atherosclerosis lesions [55].

Diabetes mellitus is a common metabolic disease characterized by a state of oxidative stress, inflammation, and endothelial dysfunction, which can lead to various complications such as ischemic heart disease, nephropathy, neuropathy, retinopathy, and impaired wound healing. LOX is critical in multiple signal transduction pathways and is involved in oxidative stress and inflammation. The findings reveal the role of LOX-1 in the development and progression of diabetic vasculopathy, which is the underlying mechanism of diabetic complications [56].

The overproduction of reactive oxygen species (ROS) damages cells and promotes inflammation, contributing to atherosclerosis and cancer. The LOX-1 receptor plays a significant role in this process by stimulating inflammation and the formation of new blood vessels. In diseases such as hypertension and diabetes, LOX-1 is involved in vascular dysfunction and the progression of cardiovascular complications. Blocking LOX-1 could be a promising strategy to treat these diseases and reduce the risk of complications.

Some potential therapeutic targets or interventions could entail the creation of specific pharmacological LOX-1 inhibitors. The development of drugs focused on blocking the activity of LOX-1 would reduce disease inflammation in conditions such as atherosclerosis and cancer.

Application of Gene Therapy: Genetic modification to regulate the expression of LOX-1 could be a strategy to control its function in the body and prevent its negative effects on cardiovascular diseases and cancer.

For example, in a phase 1 study, MEDI6570, a monoclonal antibody targeting LOX-1, was evaluated for safety and efficacy in patients with type 2 diabetes and atherosclerosis. Dose-dependent reductions in soluble LOX-1 levels were observed, with a favorable safety profile and nonlinear pharmacokinetics. MEDI6570 showed promising results in suppressing soluble LOX-1 levels and demonstrated a pharmacokinetic profile consistent with once-monthly dosing, suggesting its potential as a therapeutic option [57].

Diet and lifestyle modulation: Starting a healthy lifestyle and a diet rich in antioxidants could help reduce LOX-1 activation and its impact on cardiovascular health and cancer risk.

For example, a study investigated the effects of “Benifuuki” green tea extract, rich in O-methylated catechins, on cholesterol levels in healthy volunteers. Participants were divided into groups consuming high-dose, low-dose, or no extract for 12 weeks. Results showed a significant reduction in lectin-like oxidized LDL receptor-1 ligand-containing ApoB (LAB) levels in the high-dose group compared to the other groups. However, the groups had no significant differences in total cholesterol, triglycerides, and LDL cholesterol levels. The findings suggest that “Benifuuki” extract may help prevent arteriosclerosis by reducing LAB levels [58].

Consumption of anti-inflammatory agents: These agents could have beneficial effects by reducing the activation of LOX-1 and its contribution to diseases such as atherosclerosis and cancer.

For example, a study investigated the impact of the mitochondria-targeted antioxidant (MitoQ) on endothelial function in older adults. It was found that MitoQ treatment reduced mitochondrial reactive oxygen species (mtROS) levels, improving endothelial function and lowering circulating levels of ox-LDL. The analysis showed that exposure of endothelial cells to plasma from individuals treated with MitoQ resulted in increased nitric oxide (NO) production and decreased mtROS activity compared to the placebo. These findings suggest that MitoQ enhances endothelial function by reducing oxLDL and improving NO production. This study provides a deeper understanding of the mechanisms underlying the improvement in age-related endothelial dysfunction through MitoQ treatment [59].

These interventions have the potential to modulate LOX-1 activity and could be explored as possible therapeutic strategies in the future to treat various diseases related to this receptor.

### 4.2. Dyslipidemia as a LOX-1 Inducer

The early evolution of atherosclerotic disease centers on the accumulation of foam cells loaded with lipids. Various studies in animal models have supported this hypothesis, increasing cholesterol levels and inducing dyslipidemia in the blood. Molecules such as ICAM-1, VCAM-1, and P-selectin can accentuate the adhesion of monocytes and lymphocytes and the high content of oxidized LDL, lysophosphatidylcholine, and oxidized fatty acids in an active state of dyslipidemia induce the expression not only of these adhesion molecules but also that of sequestering receptors, such as CD-36, SR-A, and LOX-1 [60].

As shown in Figure 3, we can observe that, independently of the increase in body weight, a triggering risk factor in the development of atherosclerosis is dyslipidemia, which can be of environmental, genetic, or combined origin, initially producing an infiltration of macrophages in the subendothelium, which accumulates until reaching the early lesion; in the intermediate lesion, there is already an intra and extracellular accumulation, which gives rise to the development of the extracellular lipid nucleus or atheroma [61]. In the late stages, a fibrotic layer is produced and calcified, leading to complicated lesions, causing plaque rupture, thrombosis, and blockage of blood flow. This environment is magnified by the hyperactivation of the immune system due to the low-grade sterile inflammation that a patient in these conditions manages, as well as the increase in reactive oxygen species resulting from environmental factors and bad habits. It is important to mention the foam cells, monocytes transformed into macrophages, whose task is to recruit circulating lipids through scavenger receptors [62,63]. At the heart level, it could cause an acute myocardial infarction; at the brain level, an ischemic or hemorrhagic cerebrovascular event or a stenosis of the renal artery could cause serious damage to human health.

Likewise, dyslipidemia, rather than being a consequence, has been considered a triggering factor for diseases such as atherosclerosis and brain and cardiovascular diseases. Dyslipidemias have been grouped into those derived from environmental factors such as a high intake of lipids from the diet, sedentary lifestyle, and smoking as well as hereditary genetic disorders, which can represent a monogenic disorder (one genetic modification) and generally occur in a collaborative manner, which could represent an insignificant effect. However, when more than one genetic dyslipidemia occurs, it generates devastating consequences in subjects suffering from it [64]. 

There is a relationship between dyslipidemia and the activation of the renin–angiotensin system that triggers atherogenesis; the increasing expression of the low-density lipoprotein receptor similar to lectin LOX-1 in knockout mice for Apo-E fed a high-cholesterol diet shows extensive atherosclerosis, as determined by Sudan IV staining. In the other group, the reduction in atherosclerosis was up to 67% with antihypercholesterolemic drugs, demonstrating the relationship between the LOX-1 expression and p38 MAPK activation [65]. Experiments in rodents are mentioned later, where a reduction in hypercholesterolemia occurs through medications in conditions of high lipid intake.

In some studies, it has been shown that LOX-1 is the main receptor for ox-LDL; authors experimented with the aorta of elderly rats induced by eating a high-fat diet and found that the antihypercholesterolemic drugs administered in the control group negatively regulated the expression of ox-LDL and its receptor LOX-1 [66]. Therefore, we point to dyslipidemia as a trigger for diseases that trigger cardiovascular risk.

These data support the thesis that LOX-1 is involved in the development of cardiovascular diseases such as atherosclerosis and thrombosis and demonstrate its relationship with low-grade inflammation classified as sterile in the case of diabetes mellitus, hypertension, and metabolic syndrome in patients with dyslipidemia of exogenous or genetic origin. 

The dyslipidemia-induced expression of LOX-1 involves a series of molecular mechanisms that can be triggered by elevated levels of lipids in the bloodstream. One of the primary contributors is the presence of oxidized low-density lipoproteins and other abnormal lipids in the circulation. These abnormal lipids can directly interact with cell receptors, such as LOX-1, triggering its expression.

Various molecular mechanisms are involved in the expression of LOX-1 in dyslipidemias, with higher LDL availability coexisting with an increased risk of atherosclerosis development. The renin–angiotensin system (RAS) regulates blood pressure, water–salt balance, and cardiovascular disease pathogenesis. Angiotensin II acts as its active mediator, primarily through type 1 and type an active mediator, primarily through type 1 and 2 receptors. AT1R activation mediates most pathophysiological effects of Ang II, while the precise function of AT2R remains unclear, generally considered opposing AT1R effects. LOX-1 binds and degrades ox-LDL, linked to endothelial dysfunction, fibroblast growth, and vascular smooth muscle cell hypertrophy, crucial in atherosclerosis, hypertension, and myocardial remodeling. Evidence suggests an interaction between LOX-1 and Ang II receptors, reciprocally regulating expression and activity. Signals such as reactive oxygen species, nitric oxide, protein kinase C, and mitogen-activated protein kinases mediate this interaction, impacting dyslipidemia and RAS activation [67].

On the other hand, the interaction between oxidized LDL and its receptor, LOX-1, plays a crucial role in early-stage atherogenesis. Lysophosphatidylcholine (lyso-PC), an increased component in atherosclerotic lipoproteins, can upregulate adhesion molecules and growth factors for monocytes and T lymphocytes. This interaction highlights the significance of LOX-1 in mediating the effects of oxidized LDL in atherosclerosis development.

In summary, dyslipidemia-induced LOX-1 expression involves the interaction of ox-LDL and other abnormal lipids with specific cell receptors and the activation of intracellular signaling pathways that regulate LOX-1 gene transcription. These molecular mechanisms contribute to the development and progression of atherosclerosis in individuals with dyslipidemia [68].

It is important to emphasize the significance of recognizing monogenic and polygenic dyslipidemias in clinical practice. Monogenic dyslipidemias are rare, while common genetic variants significantly impact lipid traits under a polygenic framework. This suggests that a considerable proportion of the population with dyslipidemia may have a genetic predisposition contributing to their abnormal lipid profiles [69].

The role of small common genetic effects, especially single nucleotide polymorphisms (SNPs), in polygenic dyslipidemias emphasizes that these SNPs, while individually having a modest effect, can have a significant impact when considered collectively, especially in individuals with clinical dyslipidemia. Therefore, it highlights the importance of understanding the contribution of common genetic factors to predisposition to dyslipidemias [70].

The complexity of dyslipidemias relates to the need for an integrated approach considering genetic and environmental factors. While advances in identifying genetic variants have improved our understanding of the genetic basis of dyslipidemias, there is still much to be understood about how these variants interact with environmental factors and contribute to cardiovascular risk. Personalized medicine can benefit from a thorough assessment of genetic profiles and an understanding of how these profiles influence treatment response and prognosis in patients with dyslipidemia.

### 4.3. Expression of LOX-1 and Atherosclerosis

Atherosclerosis is classified as a chronic inflammatory disease characterized by the formation of atherosclerotic plaque in the arteries of the body, where various molecular and cellular factors are involved (accumulation of leukocytes, cells of smooth muscle, lipids, and extracellular components), causing endothelial dysfunction in the microenvironment. These factors take part in vascular pathologies such as atherosclerosis, hypertension, and diabetes in response to the induction of oxidative stress that activates LOX-1 [71].

Atherosclerosis is a multifocal silent pathology related to the immune system and the inflammation of the medium and large capillary vessels where lipids circulate, generally low-density lipoproteins, in their oxidized form. Its main characteristic is the deposition of lipids in the form of foam cells in the wall. Subendothelial differences (macrophages that recruit lipids) are due to endothelial hyperactivation in response to the damage caused by the high rate of circulating lipids in the bloodstream. The main participants in amplifying this alteration are endothelial cells, leukocytes, and intimal smooth muscle cells. Heart attacks and strokes are the most devastating consequences of atherosclerosis, which are caused primarily by the formation of the atheromatous plaque and the superimposed obstruction that causes its rupture and thrombus formation [71]. It is of the utmost importance to know what triggers luminal thrombosis after years of indolent development; if it is possible to detect it in stages before clot formation, atherosclerosis could be considered a benign disease since over 70% of coronary thrombi are fatal and are related to plaque rupture, causing coronary thrombosis in 80% of men and 60% of women [72]. In the next paragraph, we will mention the facts that are linked to the formation of foam cells. The ox-LDL-LOX-1 complex inhibits the constitutive expression of endothelial nitric oxide synthetase and contributes to the generation of ROS from cells, smooth muscle cells, and macrophages [73]. The ox-LDL-LOX-1 complex induces the expression of adhesion molecules in endothelial cells, macrophage proliferation, collagen formation, smooth muscle cell migration, and platelet activation. SR-A types 1 and II, CD-36, and LOX-1 are involved in foam cell formation through the uptake of modified LDL [74]. SR-As are normally expressed in myeloid cells but, in the presence of oxidative stress and growth factors, they are expressed in endothelial cells and smooth muscle cells [75]. 

Since 1999, the implications of LOX-1 in diseases that involve the activation of the endothelium and that stimulate the expression of factors and molecules that induce inflammation have been identified, where risk factors such as obesity, diabetes mellitus, hypertension, hypercholesterolemia, and smoking are involved. A hyper LDL-cholesterolemia can induce an overactivated endothelium, producing atheromatous lesions, referring to the renin–angiotensin system because angiotensin II stimulates the production of reactive oxygen species at the level of the capillary vessel, promoting endothelial dysfunction and the conversion of LDL to ox-LDL. Angiotensin II also induces the inflammatory response on the endothelium, stimulating the expression of molecules of adhesion and chemotactic and pro-inflammatory cytokines, which stimulates the proliferation and migration of muscle cells and also positively modulates their phenotypic change; in more advanced stages, this favors the rupture of the atherosclerotic plaque, increasing thrombogenicity in the damaged capillary vessel [55]. This supports the finding of another work, where the expression of LOX-1 was analyzed in human coronary artery endothelium, wherein an upregulation of Angiotensin II was observed; such induction triggers ox-LDL uptake [76]. This LOX-1-dependent mechanism is preceded by high glucose levels in the endothelium, which favors oxidative stress and the activation of NF-κB, activated as PKC (protein kinase C), and MAPK (mitogen-activated protein kinase) [77]. In the next paragraph, we will discuss a new term coined for patients suffering from atherosclerosis and diabetes.

A new term has been proposed: “Human Diabetic Atherosclerosis” [77,78]. This term was derived from an experiment where macrophages derived from human monocytes were stimulated with glucose (5.6 to 30 mm/L), finding that the expression of the LOX-1 gene is induced, so high glucose concentrations significantly increased the expression of LOX-1, establishing the dependence between the receptor and the formation of foam cells [77].

To verify the relationship between dyslipidemia and the expression of LOX-1 in the vessels, cDNA clones were isolated from rabbit placenta that encoded the LOX-1 homolog, which maintains a highly conserved structure, and rabbit clones were transfected into HEK-293 cells; this procedure gave them the property of binding and internalizing ox-LDL; authors reported that LOX-1 accumulated in rabbit aortas (with hereditary hyperlipidemia) compared to rabbit aortas control [79].

#### LOX-1: Relationship with Atherosclerosis and Thrombosis

Atherosclerosis is a chronic inflammatory disease that causes myocardial infarction and stroke and is linked to ischemic gangrene. Atherosclerotic plaque formation initially occurs when LDL accumulates in the intimal wall; this deposition promotes the activation of the endothelium, where leukocyte adhesion molecules and chemokines promote the recruitment of monocytes and T cells. The differentiation of monocytes to macrophages takes place, which will express pattern recognition receptors, scavenger or scavenger receptors (LOX-1), and Toll-like receptors on their membrane [80]. Macrophages undergo an additional change when the receptors garbage dump internalized lipoproteins; saturated with them, they form foam cells, accumulating in the subendothelial wall. It is, therefore, the case that Toll-like receptors transmit activation signals, which produce the expression of proinflammatory cytokines, vasoactive molecules, and proteases. In parallel, a response from T helper 1 [81] is the recognition of ant local antigens, with the participation of proinflammatory cytokines, macrophage remnants, and the participation of smooth muscle cells, producing a stable plaque, which, over time and in combination with genetic and environmental factors, increases its volume. Molecules such as collagen and elastin are added by the activation of smooth muscle cells, and the fibroatheroma is formed, slowly narrowing the passage of blood in the capillary vessel. This mechanism is silent and deadly since the plaque can pass through an unstable status where it can potentially break. Additionally, platelets recognize ligands in the eroded atherosclerotic plaque, which triggers platelet activation that, in turn, produces thrombosis to form clots that completely block the passage of blood [80,82,83].

Atheromatous encrustations are usually complicated in the upper trunk of the human body, becoming a disease at the regional level; despite this, chronic venous insufficiency in the lower limbs and its relationship with cardiovascular risk factors have been investigated, findings that are strongly associated [83]. The studies that have been carried out are aimed at analyzing the composition of the thrombotic material extracted from patients, as well as in a cohort study, in which intracoronary thrombi were aspirated from patients who had presented a myocardial infarction, where two study groups were analyzed: 32 patients with an ST-segment elevation myocardial infarction STEMI (total of coronary artery, atherosclerosis is characterized by the presence of a fibrin-rich thrombus) and 12 patients without ST-segment elevation, namely non-ST-segment elevation acute coronary syndrome NSTEMI, (incomplete coronary artery obstruction, subendocardial ischemia, and characteristically a thrombus made up of platelets), reporting that in all the samples, cholesterol crystals were detected and in 12 of 44 samples foam cells, when analyzing the immunofluorescence staining, the presence of apolipoproteins AI, apo AII, apoCIII, and apoB100 was found. Likewise, it was found that LOX-1 is mainly expressed in the thrombi of STEMI subjects than in NSTEMI patients. Two types of antibodies, sLOX-1 (soluble) and mbLOX-1 (membrane), found thrombi in patients with STEMI, that is, subjects with total coronary artery obstruction had a significantly higher relationship between the level of sLOX-1 and mbLOX-1. The opposite occurred with the NSTEMI patients; thus, the analyses carried out revealed that sLOX-1 was strongly correlated with the content of apoB100. The opposite occurred with the content of apoAI, apoAII, and apoCIII in the coronary thrombi. Additionally, it was found that apoB100 is the most abundant lipoprotein in the less electronegative subfraction of LDL (L5) and that this lipoprotein is involved in the differentiation of monocytes into macrophages, thus favoring the formation of foam cells and atheromatous plaque [84]. 

Table 1 highlights significant aspects concerning the role of the LOX-1 receptor in various pathologies, especially in atherosclerosis and cardiovascular diseases. The expression and function of this receptor have been documented in various cell types and animal models. On the other hand, the oxidation of LDL emerges as a critical factor in the development and progression of atherosclerosis, being recognized as the main ligand of LOX-1, which triggers a cascade of inflammatory effects and endothelial dysfunction. It is essential to highlight that the pathologies contemplated in the table have a direct and/or indirect interconnection with other medical conditions, such as hypertension, diabetes, and dyslipidemias, which act synergistically to enhance the development of cardiovascular diseases. In addition, studies in animal models, such as genetically modified mice and rabbits, have been used to explore the mechanisms underlying atherosclerosis and evaluate potential therapeutic interventions that could have clinical applications. Reference is also made to other target molecules, such as CD36 and SR-A, which participate with LOX-1 in lipid metabolism and the progression of these pathologies. An additional relevant implication is the close relationship between inflammation and the immune response, which plays a crucial role in the development and progression of atherosclerosis through activating the NLRP3 inflammasome and expressing pro-inflammatory cytokines. The studies collected and synthesized in this table suggest that these pathologies involve a complex interplay between genetic, environmental, and metabolic factors, endothelial dysfunction, and chronic low-grade inflammation.

The interactions between LOX-1 and specific lipoproteins, such as apoB100 and apoAI, play a crucial role in foam cell formation and the development of atherosclerotic plaques. LOX-1 is primarily responsible for binding and internalizing ox-LDL, leading to foam cell formation, a hallmark of early atherogenesis. Additionally, LOX-1 can interact with other molecules, including apoB100, contributing to the uptake of modified lipoproteins by endothelial cells and macrophages within the arterial wall [91,92].

Emerging therapeutic strategies to mitigate LOX-1-mediated atherosclerosis and thrombotic events focus on various process stages. One approach involves blocking the interaction between LOX-1 and ox-LDL using monoclonal antibodies or small molecule inhibitors. Another strategy is to modulate the expression or activity of LOX-1 through pharmacological agents or lifestyle interventions. In addition, research is directed toward interventions targeting downstream signaling pathways activated by LOX-1, such as those related to inflammation and oxidative stress.

Understanding the interactions between LOX-1 and specific lipoproteins provides insights into developing novel therapeutic approaches to combat atherosclerosis and reduce the risk of thrombotic events associated with this condition.

## 5. LOX-1 as a Biomarker

Biomarkers such as cardiac troponin and creatine kinase (CK) have been used successfully to diagnose acute myocardial infarction. However, a biomarker has not been found to identify vulnerable patients with atherosclerosis and who are thus at risk of stroke for a long time. For some years, research works have pointed to LOX-1 as a potential early biomarker of cardiovascular diseases [93]. In another study carried out on 294 subjects, three study groups were contemplated: 125 STEMI patients, 44 NSTEMI patients, and 125 patients with non-acute myocardial infarction using sLOX-1 as a biomarker with significant *p* values, the plasma levels of the said marker were maintained for 24 h after hospital admission, while other markers were not elevated and reached a notable elevation after 2 h, thus also finding the optimal cut-off value of 91.0 pg/mL of sLOX-1, differentiating STEMI patients from those with non-acute myocardial infarction, with 89.6% sensitivity and 82.4% specificity, values much higher than the biomarkers used so far such as creatine kinase-MB, cardiac troponins, myoglobin, and cardiac-type fatty acid transporter protein [94]. In another cross-sectional study with post-stroke patients (acute cerebrovascular accident), the results of 377 patients with ischemic stroke and cerebral hemorrhage were analyzed using control subjects classified by age and sex and without previous cerebrovascular history. When comparing serum levels, it was found that the values of sLOX-1 were significantly higher than in the controls, that is, an OR value of 3.80 was obtained in an ischemic cerebrovascular accident with a confidence interval of 95% and in patients with intracerebral hemorrhage, the OR was 5.97, which were independently associated through multivariable regression analysis, using a concentration greater than or equal to sLOX-1 of 1177 ng/L corresponding to the 80th percentile of patients with a cerebrovascular accident as a reference [95].

The potential use of soluble sLOX-1 as a biomarker for acute myocardial infarction and stroke presents a promising opportunity to enhance the current diagnostic landscape, which relies on established cardiac biomarkers such as cardiac troponins and creatine kinase-MB (CK-MB).

In subjects who have suffered myocardial injury as a result of coronary angioplasty or percutaneous coronary intervention, cardiac trauma can occur during or as a consequence of the procedure, resulting in a periprocedural myocardial injury (IMPP-PCI). This injury is associated with myocardial necrosis. A study with 214 patients undergoing percutaneous coronary intervention found that 33 (15.4%) developed IMPP-PCI. Patients with IMPP-PCI had higher levels of sLOX-1 than those without IMPP-PCI (167 ± 89 vs. 99 ± 68 pg/mL; *p* < 0.001). Additionally, significant correlations were found between sLOX-1 levels and troponin T, CK, and CK-MB values (r = 0.677, r = 0.682; *p* < 0.001). These findings suggest that sLOX-1 may be an early biomarker of myocardial necrosis and its ability to distinguish patients with IMPP-PCI enhances its value as a diagnostic tool in risk stratification and the development of personalized therapeutic strategies [96].

In conditions such as myocardial necrosis, especially in the early stage of acute myocardial infarction with ST-segment elevation (STEMI), the sensitivity of some important markers such as creatine kinase MB, cardiac troponins, myoglobin, and heart-type fatty acid-binding protein (H-FABP) has been analyzed. In the study by Nobuaki Kobayashi and colleagues, plasma levels of sLOX-1 were evaluated in 125 patients with STEMI, 44 with non-ST-segment elevation myocardial infarction (NSTEMI), and 125 without acute myocardial infarction (non-AMI). It was found that sLOX-1 levels were significantly higher in patients with STEMI and NSTEMI compared to those without AMI (median, 25th, and 75th percentiles: 241.0, 132.3, and 472.2 vs. 147.3, 92.9, and 262.4 vs. 64.3, 54.4 and 84.3 pg/mL, respectively). These results highlight the discriminative capacity of sLOX-1 between STEMI and non-AMI, with a sensitivity of 89.6% and a specificity of 82.4%, using an optimal cutoff value of 91.0 pg/mL. In conclusion, the findings of this study suggest that sLOX-1 may be a highly sensitive and specific biomarker for the early diagnosis of STEMI, surpassing other traditional biomarkers [94].

The most important findings are from Angela Pirillo and colleagues. The review established that sLOX-1 showed higher sensitivity and specificity than cardiac troponin T (TnT) and heart-type fatty acid binding protein (H-FABP) and could detect acute coronary syndrome (ACS) in subjects with non-significantly elevated TnT values. In ACS patients, H-FABP values correlated significantly with TnT values. In contrast, sLOX-1 values did not correlate with TnT or H-FABP, indicating that its expression is independent of the mentioned traditional markers. Additionally, plasma levels of sLOX-1 were significantly higher in subjects with acute myocardial infarction (AMI) than those without AMI. They were significantly higher in subjects with ST-segment elevation myocardial infarction than in the non-ST-segment elevation myocardial infarction group. In AMI patients, sLOX-1 levels increased in the early stages of acute myocardial infarction, persisted for 24 h after arrival at the emergency room, and declined to baseline levels 16 days after the onset of STEMI. sLOX-1 levels were elevated in patients with acute aortic dissection (AAD) and in patients with acute coronary syndrome without ST-segment elevation (NSTEACS) compared to controls. In detecting subjects with AAD and distinguishing them from those with NSTEACS, sLOX-1 proved to be a better predictor than TnT. Therefore, sLOX-1 is suggested as a sensitive and specific biomarker in diagnosing acute coronary syndromes and acute aortic dissection. It has important implications for accelerating their identification and preventing potentially fatal complications in subjects with atherosclerosis-related conditions [97].

In another study, sLOX-1 levels were analyzed in overweight/obese children and adolescents, finding that these levels were elevated during and after puberty compared to the general population. Additionally, sLOX-1 was positively associated with inflammatory markers and unfavorable cardiometabolic risk profiles, such as insulin resistance, dyslipidemia, and hypertension. These findings suggest that sLOX-1 could play an important role as an early biomarker of cardiometabolic risk and inflammation in overweight/obese children and adolescents [98].

On the other hand, patients with acute coronary syndromes (ACS) were examined to assess the role of vascular inflammation and biomarkers, specifically sLOX-1, in the development and complexity of coronary artery disease (CAD). Data were collected from patients admitted to the emergency department with unstable angina or NSTE-ACS. Elevated levels of high-sensitivity C-reactive protein (hs-CRP) and sLOX-1 were associated with more complex CAD, as per the modified Gensini score; this suggests that vascular inflammation, indicated by sLOX-1, could play a crucial role in risk prediction and ACS management, highlighting its significance as a novel predictive biomarker in individuals with this disease [99].

Incremental Value of sLOX-1 Over Established Biomarkers

Early Detection: Cardiac troponins are the gold standard for diagnosing AMI due to their high sensitivity and specificity for myocardial injury. However, troponins may only elevate several hours after the onset of AMI. sLOX-1, by contrast, might rise earlier in the course of endothelial activation and oxidative stress, potentially allowing for quicker diagnosis and intervention.Sensitivity to Subclinical Atherosclerosis: Unlike troponins and CK-MB, which increase in response to myocardial damage, sLOX-1 levels reflect endothelial dysfunction and oxidative stress, which are early events in atherosclerosis; this makes sLOX-1 a potentially valuable biomarker for identifying subclinical atherosclerosis and predicting cardiovascular events, even before structural heart damage occurs.Prognostic Value: Studies have suggested that elevated levels of sLOX-1 are associated with an increased risk of future cardiovascular events, providing prognostic information beyond the acute setting; this could be particularly useful for risk stratification and long-term management of patients with coronary artery disease.

Integrating sLOX-1 into Clinical Practice

Diagnostic Algorithms: Integrating sLOX-1 testing into routine diagnostic algorithms for AMI and stroke could complement current biomarkers, potentially leading to earlier and more accurate risk stratification and therapeutic interventions. For example, an elevated sLOX-1 level in a patient with borderline troponin levels could prompt more aggressive management or further diagnostic testing.Feasibility: Integrating sLOX-1 testing is feasible only if rapid, cost-effective, and reliable assays are developed. Current enzyme-linked immunosorbent assay (ELISA) kits for sLOX-1 are primarily used in research settings and must be adapted for routine clinical use.

Challenges and Opportunities for Implementing LOX-1 as a Routine Biomarker

Standardization: A significant challenge is the lack of standardized thresholds for sLOX-1 levels across different populations and clinical settings. Establishing these norms would require extensive validation studies.Cost and Accessibility: As with any new diagnostic tool, cost-effectiveness studies would be crucial to justify the inclusion of sLOX-1 testing in standardized diagnostic protocols, particularly in resource-limited settings.Education and Awareness: To ensure its effective use, clinicians must be educated about the implications of sLOX-1 levels and their integration into clinical decision-making processes.Regulatory Approval: Gaining regulatory approval for clinical use involves rigorous testing and validation to meet safety and efficacy standards, which can be time-consuming and costly.

Future Prospects: The potential for sLOX-1 to shortly become a routine biomarker will largely depend on ongoing and future clinical trials that can demonstrate its added value in clinical settings. With positive results, sLOX-1 could fill a crucial gap in the early diagnosis and risk stratification of cardiovascular diseases, ultimately leading to better patient outcomes.

In conclusion, it is essential to highlight the role of sLOX-1 measurements in clinical practice as they offer a window into a more precise evaluation of medical conditions related to the development and progression of diseases. Considering its potential, as found in the literature, it is highly feasible to integrate sLOX-1 testing into routine diagnostic algorithms for acute myocardial infarction (AMI) and stroke. The introduction of sLOX-1 into these algorithms could present unique challenges and opportunities. On the one hand, we may face obstacles such as additional validation of its effectiveness in different clinical contexts and standardization of testing procedures. However, the successful integration of sLOX-1 could open new doors to earlier and more accurate detection of AMI and stroke, potentially improving clinical outcomes and reducing the burden on healthcare systems. Achieving this goal will require interdisciplinary collaboration, ongoing research, and careful assessment of associated benefits and risks. According to reported findings, significant advances in incorporating sLOX-1 as a routine biomarker could be seen shortly, marking a step forward in personalized and precision healthcare with simple tests such as ELISA in patient plasma.

## 6. Concluding Remarks

We conclude that according to the evidence, the expression of LOX-1 is induced by high levels of ox-LDL that come from diet or metabolic disorders of genetic origin. As is the case with dyslipidemia, we can also point out that proinflammatory cytokines are dependent on the expression of LOX-1 due to the positive regulation of the NF-κB factor, which triggers this sterile and low-grade inflammation, related to other disorders, such as obesity, where the increase in blood lipids and pathologies such as diabetes mellitus, metabolic syndrome, and hypertension are implicit, which are related to this inflammation. This manuscript highlights elevated LOX-1 levels as an early predictor of cardiovascular risk and cerebrovascular disease in the atherosclerotic and proinflammatory environment.

## Figures and Tables

**Figure 1 ijms-25-05276-f001:**
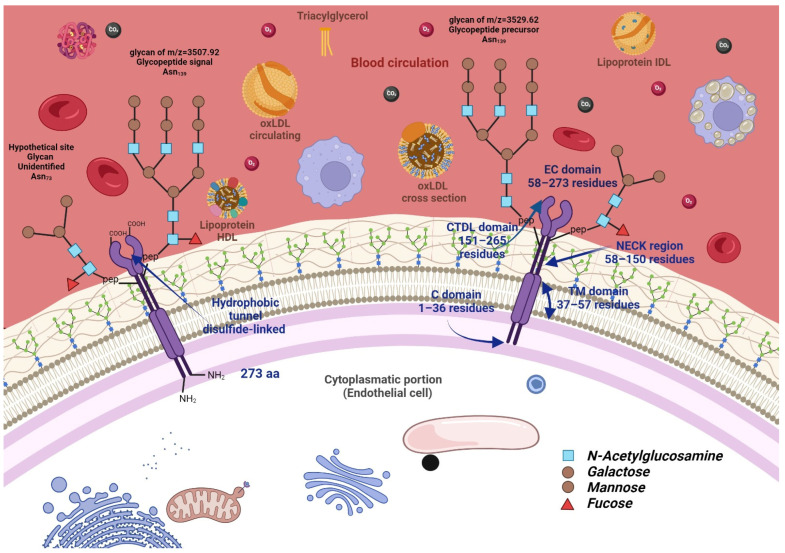
LOX-1 receptor on the cell membrane. The hypothetical structure of LOX-1 with its three potential N-Glycosylation sites. LOX-1 is a scavenger receptor type E in binding, internalizing, and recruiting ox-LDL. It is a glycoprotein, and the C-type lectin domain stands out in its structure. It interacts with the lipoprotein molecule, which comprises 131 residues, containing the extracellular coiled-coil domain (NECK) and the transmembrane domains (TM), as well as a cytoplasmic region or C-terminal lectin-like domain (CTDL), immersed in the carrier cell with its N-terminal tail comprising 34 amino acids.

**Figure 2 ijms-25-05276-f002:**
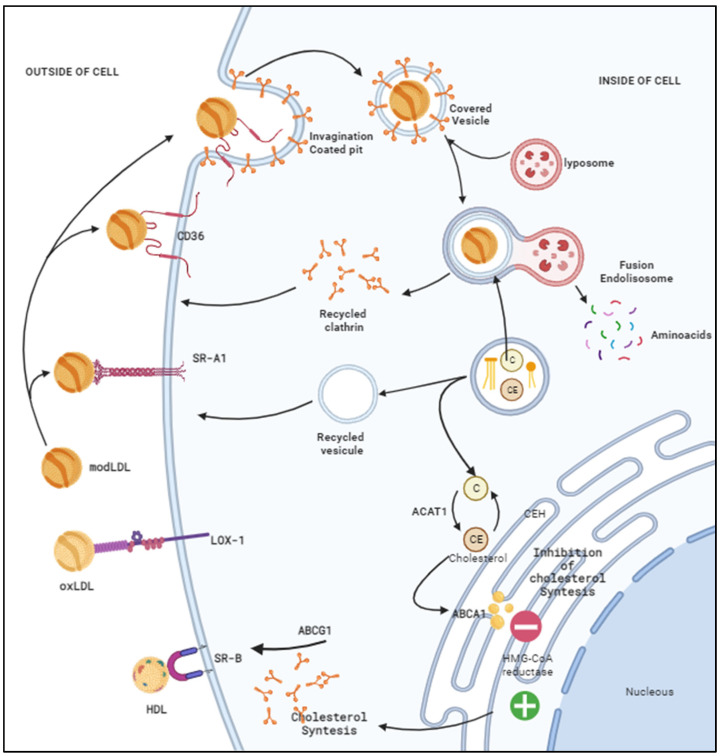
Participation of scavenger receptors in the synthesis of cholesterol. Different receptors are involved in cholesterol uptake but scavenger or scavenger receptors have mainly been identified, such as CD36, which is a receptor for collagen, and thrombospondin, which is a type B scavenger receptor and has the function of mediating; another receptor that participates in this binding with LDL is SR-A1 (scavenger receptor type A1). Another receptor is LOX-1, which binds with oxLDL and contributes to cholesterol recruitment. The binding of SR-A1 to modified LDL is related to clathrin-dependent binding (a recyclable protein involved in the formation of invaginations and intracellular vesicles) through the recognition of a cytoplasmic dileucine motif and the uptake and degradation of oxidized LDL and others. ligands. The captured cholesterol (C) is converted into esterified cholesterol (EC) by acylcholesterolacyltransferase (ACAT1), which is the beginning of the formation of the foam cell. On the other hand, ABCG1 (ATP binding cassette subfamily G member 1) is a cholesterol carrier whose function is to reach the cell surface to promote the release of cholesterol through the high-density lipoprotein HDL, which is in turn captured by the scavenger receptor class B type 1 (SR-B1), both allowing the release of cholesterol to the plasma membrane to maintain cellular homeostasis. Intracellular free and esterified cholesterol concentrations are well regulated by the enzyme hydroxymethylglutaryl-CoA reductase (HMG-CoA reductase), which is in turn regulated by ACAT1. On the other hand, cholesterol ester hydrolase (CEH) hydrolyzes cholesterol esters to cholesterol and free fatty acids. Both CEH and ACAT1 are the enzymes responsible for regulating cholesterol and esterified cholesterol levels. Adapted from “Role of lipin-1 unmodified-LDL Induced Pro-inflammatory Response”, by BioRender.com (2023). Retrieved from https://app.biorender.com/biorender-templates (accessed on 2 February 2024).

**Figure 3 ijms-25-05276-f003:**
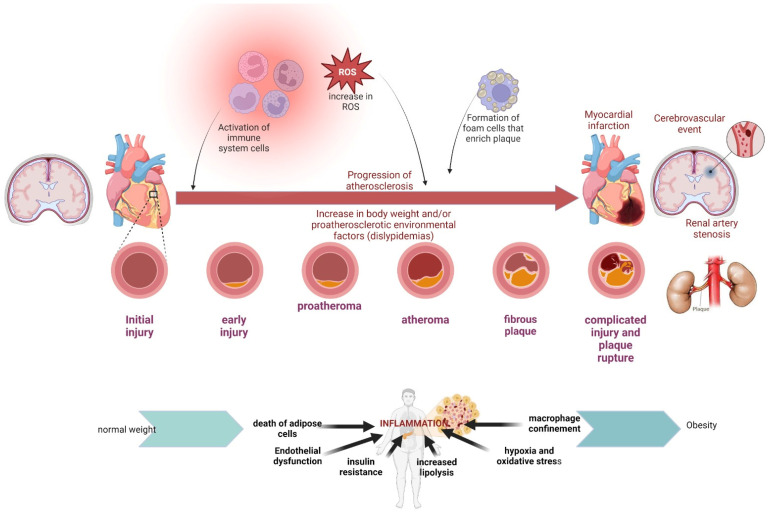
Factors that increase the risk of atherosclerosis. The factors that increase the risk of developing atherosclerosis are mainly of environmental origin and that can be modifiable; they are those that produce chronic sterile or low-grade inflammation, such as increased body weight, increased levels of low-grade lipoproteins, blood density, coupled with the increase in reactive oxygen species (poor diet, sedentary lifestyle, and tobacco consumption), promoting the death of adipose cells, endothelial dysfunction, insulin resistance, increased lipolysis, hypoxia, and oxidative stress; as well as the recruitment of macrophages due to the activation of the cells of the immune system (transformed monocytes), these phages will consume excess cholesterol to deposit on the arterial walls, producing lesions and proatheromas in early stages that, in advanced stages, can become complicated, producing a rupture of plaque; this fact can be fatal, since, by obstructing the lumen of the blood vessel in the arteries of the heart, it can cause heart attacks or brain vascular accident (ischemic or hemorrhagic). Another non-modifiable factor that increases the risk of developing atherosclerosis is genetic dyslipidemia. Adapted from “Atherosclerosis Progression”, by BioRender.com (2023). Retrieved from https://app.biorender.com/biorender-templates (accessed on 2 February 2024).

**Table 1 ijms-25-05276-t001:** Pathological function of LOX-1.

Pathology	Target Molecule	Modelo Used	Outcomes
Atherogenesis/Atherosclerosis	Lectin-like-ox-LDL receptor-1(LOX-1)	In vitro vascular smooth muscle (SMC) cultures and human coronary artery and aortic samples from WHHL rabbits with atherosclerotic lesions for in vivo studies	Ref. [33] Proinflammatory cytokines and LOX-1—Upregulated LOX-1 is a response to proinflammatory cytokines and PPARgamma can mitigate the proinflammatory effects of these cytokines
	Not applicable (Review)	Not applicable (Review)	Ref. [34] Proinflammatory cytokines and LOX-1—Endothelial dysfunction, macrophage migration, and VSMC proliferation
	LOX-1	Watanabe heritable hyperlipidemic (WHHL) rabbits	Ref. [79] Expression of LOX-1 and atherosclerosis—Enhanced expression of LOX-1 produced in endothelial cells of the intimal layer in early atherosclerosis
Atherosclerosis	Genes related to cholesterol and lipid metabolism: LOX-1 and gen APOE	Knockout mice (KO) for Apoe and Ldlr genes. Apoe KO mice: Develop hypercholesterolemia and atherosclerotic lesions at an early age. Ldlr KO mice: They have a similar phenotype but are more attenuated	Ref. [85] Expression of LOX-1 and atherosclerosis—The collection of accumulated mutations over the past decades represents a valuable source of animal models for studying human genetic disorders
	LOX-1	Reviews previous studies investigating the expression of LOX-1 in vascular endothelial cells and macrophages in human atherosclerotic lesions and in animal models (not specified)	Ref. [41] Proinflammatory inductors—Controlling gene expression and targeting specific cells offers hope for developing new treatments to regress atherosclerotic lesions and potentially prevent their formation
	LOX-1 and ox-LDL (low-density lipoprotein)	Macrophages of LOX-1 knockout mice	Ref. [38] Proinflammatory inductors—Lysophosphatidylcholine and proinflammatory cytokines highly induce the expression of LOX-1, which increases the uptake of ox-LDL by macrophages in atherosclerotic lesions
	Upregulation of LOX-1 by palmitic acid stimulation	Macrophages THP1 y Raw264.7	Ref. [39] Proinflammatory inductors—Palmitic acid promotes the expression of LOX-1 in macrophages, contributing to the development of atherosclerosis by increasing ox-LDL uptake
	LDL	Review of Atherosclerosis. Endothelial cells, leukocytes, and intimal smooth muscle cells	Ref. [86] Proinflammatory inductors— Atherosclerosis is a multifocal immunoinflammatory disease of medium and large arteries involving endothelial cells, leukocytes, and smooth muscle cells
	Scavenger receptor A and CD36 and LOX-1	Not applicable (Review)	Ref. [73] Relationship with atherosclerosis and thrombosis—The primary contributors to the accumulation of lipids and inflammatory cells in arterial walls are CD36 and LOX-1, playing a crucial role in plaque genesis and progression
	LOX-1	Human coronary artery endothelial cells	Ref. [76] Relationship with atherosclerosis and thrombosis—The interaction between oxidized ox-LDL and angiotensin II increases ox-LDL uptake by coronary artery endothelial cells, leading to enhanced cell injury
	LOX-1	Apo-E knockout mice fed with a high-cholesterol diet	Ref. [65] Dyslipidaemia as a LOX-1 inducer— The synergistic inhibitors of LOX-1 (rosuvastatin and candesartan) affected the expression and phosphorylation of MAPK, reducing atherosclerosis by 67% in the murine model
	Scavenger receptors and Toll-like receptors	Inflammatory and atherosclerotic processes in humans (Review)	Ref. [80] Relationship with atherosclerosis and thrombosis—Atherosclerosis is the cause of heart attacks, strokes, and ischemic gangrene. This condition involves the recruitment of transformed monocytes and the activation of T cells. In later stages, it leads to plaque formation and rupture
	Components of LDL carrying cholesterol	Mouse models (Review)	Ref. [81] Targeted deletion of genes encoding costimulatory factors and proinflammatory cytokines results in less disease in animal models
	Scavenger receptors and Toll-like receptors	Not applicable (Review)	Ref. [82] Relationship with atherosclerosis and thrombosis—Atherosclerosis begins with the internalization of ox-LDL in the subendothelium, increasing its permeability and the expression of cytokines, chemokines, and adhesion molecules
Atherosclerosis	LOX-1 and global DNA methylation.	Apolipoprotein-E-deficient (ApoE−/−) mice treated to induce hyperhomocysteinemia (HHcy)	Ref. [47] Proinflammatory inductors— Homocysteine-induced atherosclerosis is closely associated with induced hypomethylation status in blood vessels and this process is partially mediated by LOX-1 DNA methylation. Homocysteine (Hcy) is an independent risk factor of atherosclerosis but is involvement with the methionine cycle
Hypertension and atherogenesis	LOX-1 and ox-LDL	Review of studies of cultures of vascular endothelial cells and animal models of atherosclerosis and hyperlipidemia, such as WHHL rabbits and SHR rats, cholesterol-fed mice, and monkeys	Ref. [55] Expression of LOX-1 and atherosclerosis—Ang II functions to enhance ox-LDL uptake, amplifying cell injury through the overexpression of LOX-1 in the endothelium
Endothelial dysfunction and atherogenesis	LOX-1 Membrane type 1 MMP (MT1-MMP) Integral role in RhoA and Rac1-dependent signaling pathways (ox-LDL)	In human aortic endothelial cells (HAECs) in culture and in Watanabe rabbits with heritable hyperlipidemia. Fluorescent immunostaining and immunoprecipitation to analyze the colocalization and complex formation between lectin-like-ox-LDL receptor-1(LOX-1) and MT1-MMP in HAECs	Ref. [87] Expression of LOX-1 and atherosclerosis—The main focus of the LOX-1-MT1-MMP axis in the activation pathways of RhoA and Rac1 in response to ox-LDL is highlighted, positioning LOX-1 as a potential therapeutic target for endothelial dysfunction
Atherosclerosis and endothelial dysfunction	LOX-1, NLRP3 inflammasome, and DNase II.	THP-1 macrophages, human monocytic cell line, and primary peritoneal macrophages from C57BL/6 mice	Ref. [35] Pathological function of LOX-1- LOX-1 and mitochondrial DNA damage drive cardiovascular diseases like atherosclerosis. In macrophages, whereby LOX-1 induces ROS, autophagy, and NLRP3 inflammasome, are crucial in inflammation
	LOX-1	Human aortic endothelial cells (HAECs) are used and in vitro experiments are performed to study the effect of CRP on the expression of LECTIN-LIKE OX-LDL RECEPTOR-1 (LOX-1), the adhesion of monocytes to endothelial cells, and the uptake of ox-LDL by endothelial cells	Ref. [40] Proinflammatory inductors—C-reactive protein is an inflammatory marker that predicts cardiovascular events, promotes endothelial dysfunction, and increases the expression of endothelial LOX-1
Coronary heart disease and Atherosclerosis	LDL An oxidative modification The acetyl LDL or scavenger receptor	Review of intervention studies using probucol as an antioxidant in the LDL receptor-deficient animal model for atherosclerosis (WHHL rabbit)	Ref. [42] Proinflammatory inductors— Hypercholesterolemia is one of the factors triggering coronary artery disease due to its significance. It is important to note that this disease is multifactorial
Atherosclerosis, cardiovascular disease	High-density lipoprotein (HDL).	Human endothelial cells to evaluate the ability of 15-LO-modified HDL3 to inhibit the expression of TNF-α-mediated MCP-1 and adhesion molecules. In addition, assays were performed to evaluate the adhesion of monocytes to endothelial cells exposed to HDL3 modified by 15-LO. The pathways responsible for the effects induced by 15-LO-modified HDL3 were investigated, focusing mainly on the activation of NF-κB and AP-1	Ref. [43] Proinflammatory inductors—Endothelial dysfunction, one of the earliest events in vascular atherogenesis, is exacerbated by the enzymatic modification of HDL3 by 15-LO, resulting in a dysfunctional lipoprotein with proinflammatory characteristics
Atherosclerosis accelerated by diabetes mellitus	LDL glycosylated	Normal human fibroblasts. The LDL glycosylated in vitro in a slow non-enzymatic reaction obtained from plasma from normolipidemic subjects	Ref. [37] Proinflammatory inductors—Glycosylation of LDL significantly impairs its cellular interactions, including uptake and degradation by fibroblasts, and its ability to stimulate cholesteryl ester synthesis, which may contribute to accelerated atherosclerosis in diabetic patients
Atherosclerosis and cardiovascular disease (ACVD)	“Find-Me” signals (such as lysophosphatidylcholine), “Eat-Me” signals (such as phosphatidylserine, Mer tyrosine kinase receptor (MerTK), and milk fat globule-EGF factor 8), and “Don’t Eat-Me” signals (such as cluster of differentiation 47 (CD47)).	Studies on atherosclerotic plaques and specialized cells like macrophages and dendritic cells, as well as non-specialized cells with phagocytic capability like endothelial cells and smooth muscle cells. Additionally, the use of Watanabe heritable hyperlipidaemic rabbits (WHHL) as an animal model (Review)	Ref. [60] Dyslipidaemia as a lox-1 inducer—the impaired process of efferocytosis involving various cell types and signalling molecules contributes significantly to the progression of atherosclerosis, highlighting its potential as a target for personalized treatment of cardiovascular disease
Endothelial dysfunction	LOX-1	Review in microvascular endothelial cells (HMEC-1)	Ref. [48] Proinflammatory inductors—LOX-1 acts as a mediator of endothelial dysfunction and, in turn, as a promoter of ROS generation, suggesting the involvement of signaling pathways such as mitogen-activated protein kinases (MAPKs)
	LOX-1	Review of studies in vitro and in animal models (rats and human coronary artery endothelial cells)	Ref. [88] Expression of LOX-1 and atherosclerosis- LOX-1 contributes to endothelial dysfunction by inducing functional changes in these cells, impacting vascular homeostasis. It triggers ROS generation and reduces NO release
Coronary heart disease	Not specific (Review)	Mouse models with diabetes-accelerated atherosclerosis	Ref. [78] Expression of LOX-1 and atherosclerosis— The review mentions that the mouse models currently used to study diabetic atherosclerosis must meet the following criteria: persistent diabetes, acceleration of atherosclerosis, and resistance to medical intervention, the most common of which are classified according to the type of diabetes
Diabetic atherosclerosis	Advanced glycation endproducts–BSA induced LOX-1 expression). VLDL/LDL prominently increased LOX-1	Streptozotocin-induced diabetic rats(aorta) Cultured aortic endothelial cells stimulated by diabetic rat serum	Ref. [46] Proinflammatory inductors—The serum of diabetic rats shows an accumulation of LOX-1 ligand activity, mainly in the VLDL/LDL fractions
	ox-LDL	Human monocyte-derived macrophage (MDM) The ox-LDL, hyperglycemia, glycoxidation, lipoxidation, increased oxidative stress, and activation of protein kinase C (PKC)	Ref. [77] Proinflammatory cytokines and LOX-1—The most important findings are the glucose-dependent positive regulation of LOX-1 expression and its association with atherosclerosis progression. High glucose involves several signalling pathways, including protein kinase C, mitogen-activated protein kinases, nuclear factor κB, and activated protein-1. The role of oxidative stress is directly related to foam cell formation promoted by high glucose. In summary, hyperglycemia contributes to the development of atherosclerosis in type 2 diabetes
Cardiovascular disease	Guanosine triphosphatase small Rho and its target Rho-kinase	Not applicable (Review)	Ref. [89] Pathological function of LOX-1—The activation of the Rho/Rho-kinase signaling pathway plays a crucial role in regulating blood pressure and vasoconstriction, with significant implications in cardiovascular diseases such as hypertension and atherosclerosis.
Endothelial dysfunction	Rac1 (a member of the GTPase family of the Rho family)	Not applicable (Review)	Ref. [44] The pathological function of LOX-1—The main role of Rac1 as a regulator of multiple cellular signalling pathways affecting cytoskeleton organization, transcription, and cell proliferation, which have implications in endothelial dysfunction and associated pathological conditions such as tumorigenesis, neurodegenerative disorders, liver cirrhosis, and cardiovascular remodeling/hypertension, has been investigated
	LOX-1 and Nod-like receptor nucleotide-binding domain leucine rich repeat containing protein 3 (NLRP3).	A cellular model of endothelial progenitor cel (EPCs) treated with different concentrations of astragaloside IV (ASV) and ox-LDL as a stimulus for cellular dysfunction	Ref. [51] Proinflammatory inductors—Astragaloside IV (ASV) demonstrates protective effects against oxidized low-density lipoprotein (ox-LDL)-induced dysfunction in endothelial progenitor cells (EPCs) through modulation of the LOX-1/NLRP3 pathway, highlighting its potential therapeutic role in diabetic vascular complications
	LOX-1 and Nod-like receptor nucleotide-binding domain leucine-rich repeat-containing protein 3 (NLRP3).	This is a cellular model of endothelial progenitor cells (EPCs) treated with different concentrations of astragaloside IV (ASV) and ox-LDL as a stimulus for cellular dysfunction	Ref. [51] Proinflammatory inductors—Astragaloside IV (ASV) demonstrates protective effects against oxidized low-density lipoprotein (ox-LDL)-induced dysfunction in endothelial progenitor cells (EPCs) through modulation of the LOX-1/NLRP3 pathway, highlighting its potential therapeutic role in diabetic vascular complications
Hypertension	LOX-1	Spontaneous hypertensive rats (SHR-SP)—Wistar Kyoto Rats (WKY), Dahl salt-sensitive (DS), and salt-resistant rats (DR)	Ref. [36] The pathological function of LOX-1—The increased expression of the endothelial receptor for oxidized LDL (LOX-1) in hypertensive rats suggests the involvement of LOX-1 in endothelium-dependent vasodilation dysfunction
	No specific	Spontaneously hypertensive rat (SHR) and the normotensive Wistar Kyoto (WKY) control	Ref. [90] Proinflammatory inductors—The research on LOX-1 expression in spontaneously hypertensive rats (SHR) offers an invaluable opportunity to better understand the relationship between hypertension and endothelial dysfunction, opening new perspectives in cardiovascular disease pathogenesis research
	LOX-1	A molecular–cellular interaction model was reviewed to study the relationship between LDL oxidation ox-LDL, LOX-1, angiotensin II (Ang II), and Ang II receptor type 1 (AT1R) in the pathogenesis of hypertension	Ref. [45] Proinflammatory inductors—Hypertension is associated with vascular oxidative stress, which promotes proliferation and hypertrophy of vascular smooth muscle cells, collagen deposition, endothelial dysfunction, altered vasoconstriction, increased levels of ox-LDL, and changes in the renin–angiotensin system and angiotensin II levels
Aortic aneurysm (AA)	Peptide p210 related to apoB-100.	Apolipoprotein E-deficient (ApoE−/−) mice were treated with a vaccine utilizing peptide p210 and then implanted with a pump releasing angiotensin II (Ang II) to induce AA	Ref. [72] Expression of lox-1 and atherosclerosis— The role of apoB-containing lipoproteins in atherosclerosis was studied and suggests a potential connection with protection against abdominal aortic aneurysm through an apoB-related vaccine in mice
Not specified	LOX-1	Atherosclerosis-susceptible C57BL/6 and atherosclerosis-resistant C3H/HeJ mice fed an atherogenic high-fat diet. (Murine macrosialin) MS does not function as an ox-LDL receptor	Ref. [74] Expression of lox-1 and atherosclerosis—The study shows that MS does not act as a receptor for ox-LDL on the cell surface, even though its expression increases in response to a high-fat diet and oxLDL treatment
Chronic Venous Disease (CVD)	No specific	An observational study was conducted on 902 women from the general population, aged 45 to 54 years Evaluation of cardiovascular risk factors (ankle/brachial systolic blood pressure index (ABI) and carotid intima-media thickness of the common carotid arteries measured by ultrasound)	Ref. [83] Relationship with atherosclerosis and thrombosis—Soluble LOX-1 levels are elevated in acute myocardial infarction, suggesting a role for the more electronegative fraction (L5) of LDL, which is associated with promoting atherogenesis and thrombus formation
Coronary Thrombosis	LOX-1	This is an observational study that analyzed aspirated coronary thrombi from patients with STEMI and without ST-segment elevation myocardial infarction (NSTEMI) to evaluate the expression of LECTIN-LIKE OX-LDL RECEPTOR-1 (LOX-1) and the relationship between sLECTIN-LIKE OX-LDL RECEPTOR-1 (LOX-1) and L5 levels in peripheral blood	Ref. [84] Relationship with atherosclerosis and thrombosis—sLOX-1 levels are elevated in acute myocardial infarction (AMI), particularly in ST-segment-elevation myocardial infarction (STEMI), suggesting a potential role for L5 in triggering atherogenesis and promoting thrombi formation
Hypercholesterolemia	LOX-1	Wistar rats were divided into groups; some were fed a high-cholesterol diet and others were treated with Aegeline/atorvastatin along with the same diet.	Ref. [66] Dyslipidemia as a LOX-1 inducer— Aegeline (AG) is shown to be an effective anti-hypercholesterolemic agent by reducing Ox-LDL and LOX-1 levels, suggesting its potential as a novel diagnostic strategy for hypercholesterolemia and vascular diseases

## Data Availability

Not applicable.

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
