# Peer review of "LOX-1 in Cardiovascular Disease: A Comprehensive Molecular and Clinical Review"

_ijms, 2024, doi:10.3390/ijms25105276_

Round 1
Reviewer 1 Report
Comments and Suggestions for Authors
The manuscript entitled Transmembrane glycoprotein LOX -1 (Lectin-Like Oxidized Low-Density Lipoprotein Receptor 1) and its physiological and pathophysiological functions is a narrative review about Lectin-Like Oxidized Low-Density Lipoprotein Receptor 1 (LOX-1) to support the detection diagnosis, and prevention of various pathologies such as obesity, diabetes mellitus, metabolic syndrome, and hypertension. The authors underlined the role of elevated LOX-1 levels as an early predictor of cardiovascular risk and cerebrovascular disease in the atherosclerotic and proinflammatory environment.
General comments
The narrative review is well structured and well written. The molecular characteristics of LOX-1 contribute to a better understanding of this receptor for oxidized low-density lipoproteins in atherosclerosis. Physiological function of LOX-1 is synthetized in figure 2, which is another important step for understanding the role of elevated LOX-1 levels in atherosclerosis, especially since this chapter makes the transition to pathology (chapter 4). This chapter (number 4) is the mean part of this narrative review. I suggest synthesizing this chapter in a table with all-important studies for each sub-chapter.
This narrative review is important as stage of knowledge for PhD students who want study this theme therefore it deserve to be published.
Specific comments
There are many acronyms. Please verify if all these acronyms are necessary and are well used. For example: CVD and DM2 are used only for two times.
There are some perspectives as biomarker for LOX-1? Please discuss this.
Author Response
We thank the reviewer for the excellent comments on our manuscript. Below, we describe how we have included the comments:
- The narrative review is well structured and well written. The molecular characteristics of LOX-1 contribute to a better understanding of this receptor for oxidized low-density lipoproteins in atherosclerosis. Physiological function of LOX-1 is synthetized in figure 2, which is another important step for understanding the role of elevated LOX-1 levels in atherosclerosis, especially since this chapter makes the transition to pathology (chapter 4). This chapter (number 4) is the mean part of this narrative review. I suggest synthesizing this chapter in a table with all-important studies for each sub-chapter.
R// We included the table 1:
Table 1. Pathological function of LOX-1
Pathology |
Target molecule |
Modelo used |
Outcomes |
Atherogenesis/Atherosclerosis |
Lectin-like-ox-LDL receptor-1(LOX-1) |
In vitro vascular smooth muscle (SMC) cultures and human coronary artery and aortic samples from WHHL rabbits with atherosclerotic lesions for in vivo studies.
|
[36] Proinflammatory cytokines and LOX-1 - Upregulated LOX-1 is a response to proinflammatory cytokines, and PPARgamma can mitigate the proinflammatory effects of these cytokines. |
Not applicable (Review) |
Not applicable (Review) |
[43] Proinflammatory cytokines and LOX-1 - Endothelial dysfunction, macrophage migration, and VSMC proliferation. |
|
LOX-1 |
Watanabe heritable hyperlipidemic (WHHL) rabbits, |
[73] Expression of LOX-1 and atherosclerosis - Enhanced expression of LOX-1 produced in endothelial cells of the intimal layer in early atherosclerosis.
|
|
Atherosclerosis
|
Genes related to cholesterol and lipid metabolism: LOX-1 and gen APOE |
Knockout mice (KO) for Apoe and Ldlr genes. Apoe KO mice: Develop hypercholesterolemia and atherosclerotic lesions at an early age. Ldlr KO mice: They have a similar phenotype but are more attenuated.
|
[79] Expression of LOX-1 and atherosclerosis -The collection of accumulated mutations over the past decades represents a valuable source of animal models for studying human genetic disorders |
LOX-1 |
Reviews previous studies investigating the expression of LOX-1 in vascular endothelial cells and macrophages in human atherosclerotic lesions and in animal models (not specified). |
[45] Proinflammatory inductors - Controlling gene expression and targeting specific cells offers hope for developing new treatments to regress atherosclerotic lesions and potentially prevent their formation |
|
LOX-1 and ox-LDL (low-density lipoprotein) |
Macrophages of LOX-1 knockout mice |
[49] Proinflammatory inductors—Lysophosphatidylcholine and proinflammatory cytokines highly induce the expression of LOX-1, which increases the uptake of ox-LDL by macrophages in atherosclerotic lesions. |
|
Upregulation of LOX-1 by palmitic acid stimulation |
Macrophages THP1 y Raw264.7 |
[50] Proinflammatory inductors - Palmitic acid promotes the expression of LOX-1 in macrophages, contributing to the development of atherosclerosis by increasing ox-LDL uptake." |
|
LDL |
Review of Atherosclerosis. Endothelial cells, leukocytes, and intimal smooth muscle cells. |
[80] Proinflammatory inductors - Atherosclerosis is a multifocal immunoinflammatory disease of medium and large arteries involving endothelial cells, leukocytes, and smooth muscle cells. |
|
Scavenger receptor A and CD36 and LOX-1 |
Not applicable (Review) |
[66] Relationship with atherosclerosis and thrombosis - The primary contributors to the accumulation of lipids and inflammatory cells in arterial walls are CD36 and LOX-1, playing a crucial role in plaque genesis and progression |
|
LOX-1
|
Human coronary artery endothelial cells (HCAECs). |
[70] Relationship with atherosclerosis and thrombosis- The interaction between oxidized ox-LDL and angiotensin II increases ox-LDL uptake by coronary artery endothelial cells, leading to enhanced cell injury. |
|
LOX-1
|
Apo-E knockout mice fed with a high-cholesterol diet |
[62] Dyslipidaemia as LOX-1 inducer - The synergistic inhibitors of LOX-1 (rosuvastatin and candesartan) affected the expression and phosphorylation of MAPK, reducing atherosclerosis by 67% in the murine model. |
|
Scavenger receptors and Toll-like receptors
|
Inflammatory and atherosclerotic processes in humans. (Review) |
[74] Relationship with atherosclerosis and thrombosis - Atherosclerosis is the cause of heart attacks, strokes, and ischemic gangrene. This condition involves the recruitment of transformed monocytes and the activation of T cells. In later stages, it leads to plaque formation and rupture. |
|
Components of LDL carrying cholesterol
|
Mouse models (Review) |
[75] 4.3.1- Targeted deletion of genes encoding costimulatory factors and proinflammatory cytokines results in less disease in animal models. |
|
Scavenger receptors and Toll-like receptors |
Not applicable (Review) |
[76] Relationship with atherosclerosis and thrombosis - Atherosclerosis begins with the internalization of ox-LDL in the subendothelium, increasing its permeability and the expression of cytokines, chemokines, and adhesion molecules. |
|
Atherosclerosis |
LOX-1 and global DNA methylation. |
Apolipoprotein-E-deficient (ApoE-/-) mice treated to induce hyperhomocysteinemia (HHcy). |
[53] Proinflammatory inductors - Homocysteine-induced atherosclerosis is closely associated with induced hypomethylation status in blood vessels, and this process is partially mediated by LOX-1 DNA methylation. Homocysteine (Hcy) is an independent risk factor of atherosclerosis, but its involvement with the methionine cycle) |
Hypertension and atherogenesis |
LOX-1 and ox-LDL |
Review of studies of Cultures of vascular endothelial cells and animal models of atherosclerosis and hyperlipidemia, such as WHHL rabbits and SHR rats, Cholesterol-fed mice, and monkeys |
[69] Expression of LOX-1 and atherosclerosis - Ang II functions to enhance ox-LDL uptake, amplifying cell injury through the overexpression of LOX-1 in the endothelium.
|
Endothelial dysfunction and atherogenesis.
|
LOX-1 Membrane type 1 MMP (MT1-MMP) Integral role in RhoA and Rac1-dependent signaling pathways (ox-LDL)
|
In human aortic endothelial cells (HAECs) in culture, in Watanabe rabbits with heritable hyperlipidemia. Fluorescent immunostaining and immunoprecipitation to analyze the colocalization and complex formation between Lectin-like-ox-LDL receptor-1(LOX-1) and MT1-MMP in HAECs. |
[41] Expression of LOX-1 and atherosclerosis - The main focus of the LOX-1-MT1-MMP axis in the activation pathways of RhoA and Rac1 in response to ox-LDL is highlighted, positioning LOX-1 as a potential therapeutic target for endothelial dysfunction. |
Atherosclerosis and endothelial dysfunction |
LOX-1, NLRP3 inflammasome, and DNase II. |
THP-1 macrophages, human monocytic cell line, and primary peritoneal macrophages from C57BL/6 mice
|
[38] Pathological function of LOX-1- LOX-1 and mitochondrial DNA damage drive cardiovascular diseases like atherosclerosis. In macrophages, LOX-1 induces ROS, autophagy, and NLRP3 inflammasome, crucial in inflammation.
|
LOX-1 |
Human aortic endothelial cells (HAECs) are used, and in vitro experiments are performed to study the effect of CRP on the expression of LECTIN-LIKE OX-LDL RECEPTOR-1 (LOX-1), the adhesion of monocytes to endothelial cells, and the uptake of ox-LDL by endothelial cells. |
[44] Proinflammatory inductors - C-reactive protein is an inflammatory marker that predicts cardiovascular events, promotes endothelial dysfunction, and increases the expression of endothelial LOX-1. |
|
Coronary heart disease and Atherosclerosis |
LDL An oxidative modification The acetyl LDL or scavenger receptor |
Review of intervention studies using probucol as an antioxidant in the LDL receptor-deficient animal model for atherosclerosis (WHHL rabbit). |
[46] Proinflammatory inductors - Hypercholesterolemia is one of the factors triggering coronary artery disease due to its significance. It's important to note that this disease is multifactorial. |
Atherosclerosis, cardiovascular disease.
|
High-density lipoprotein (HDL).
|
Human endothelial cells to evaluate the ability of 15-LO-modified HDL3 to inhibit the expression of TNF-α-mediated MCP-1 and adhesion molecules. In addition, assays were performed to evaluate the adhesion of monocytes to endothelial cells exposed to HDL3 modified by 15-LO. The pathways responsible for the effects induced by 15-LO-modified HDL3 were investigated, focusing mainly on the activation of NF-κB and AP-1. |
[47] Proinflammatory inductors—Endothelial dysfunction, one of the earliest events in vascular atherogenesis, is exacerbated by the enzymatic modification of HDL3 by 15-LO, resulting in a dysfunctional lipoprotein with proinflammatory characteristics. |
Atherosclerosis accelerated by diabetes mellitus. |
LDL glycosylated |
normal human fibroblasts. The LDL glycosylated in vitro in a slow non-enzymatic reaction obtained from plasma from normolipidemic subjects. |
[48] Proinflammatory inductors - Glycosylation of LDL significantly impairs its cellular interactions, including uptake and degradation by fibroblasts, and its ability to stimulate cholesteryl ester synthesis, which may contribute to accelerated atherosclerosis in diabetic patients |
Atherosclerosis and cardiovascular disease (ACVD). |
"Find-Me" signals (such as lysophosphatidylcholine), "Eat-Me" signals (such as phosphatidylserine, Mer tyrosine kinase receptor (MerTK), and milk fat globule-EGF factor 8), and "Don't Eat-Me" signals (such as cluster of differentiation 47 (CD47)). |
Studies on atherosclerotic plaques and specialized cells like macrophages and dendritic cells, as well as non-specialized cells with phagocytic capability like endothelial cells and smooth muscle cells. Additionally, the use of Watanabe heritable hyperlipidaemic rabbits (WHHL) as an animal model (Review) |
[60] Dyslipidaemia as lox-1 inducer - the impaired process of efferocytosis involving various cell types and signalling molecules, contributes significantly to the progression of atherosclerosis, highlighting its potential as a target for personalized treatment of cardiovascular disease |
Endothelial dysfunction |
LOX-1 |
Review in Microvascular endothelial cells (HMEC-1). |
[54] Proinflammatory inductors - LOX-1 acts as a mediator of endothelial dysfunction and, in turn, as a promoter of ROS generation, suggesting the involvement of signaling pathways such as mitogen-activated protein kinases (MAPKs) |
LOX-1 |
Review of studies in vitro and in animal models (rats and human coronary artery endothelial cells).
|
[64] Expression of LOX-1 and atherosclerosis- LOX-1 contributes to endothelial dysfunction by inducing functional changes in these cells, impacting vascular homeostasis. It triggers ROS generation and reduces NO release |
|
Coronary heart disease
|
Not specific (Review) |
Mouse models with diabetes-accelerated atherosclerosis. |
[72]Expression of LOX-1 and atherosclerosis - The review mentions that the mouse models currently used to study diabetic atherosclerosis must meet the criteria: persistent diabetes, acceleration of atherosclerosis, and resistance to medical intervention, the most common of which are classified according to the type of diabetes. |
Diabetic atherosclerosis. |
Advanced glycation endproducts–BSA induced LOX-1 expression). VLDL/LDL prominently increased LOX-1 |
Streptozotocin-induced diabetic rats(aorta). cultured aortic endothelial cells stimulated by diabetic rat serum. |
[52] Proinflammatory inductors - The serum of diabetic rats shows an accumulation of LOX-1 ligand activity, mainly in the VLDL/LDL fractions. |
ox-LDL |
human monocyte-derived macrophage (MDM) The ox-LDL, hyperglycemia and (glycoxidation and lipoxidation, increased oxidative stress, and activation of protein kinase C (PKC) |
[71] Proinflammatory cytokines and LOX-1—The most important findings are the glucose-dependent positive regulation of LOX-1 expression and its association with atherosclerosis progression. High glucose involves several signalling pathways, including protein kinase C, mitogen-activated protein kinases, nuclear factor κB, and activated protein-1. The role of oxidative stress is directly related to foam cell formation promoted by high glucose. In summary, hyperglycemia contributes to the development of atherosclerosis in type 2 diabetes. |
|
Cardiovascular disease |
Guanosine triphosphatase small Rho and its target Rho-kinase |
Not applicable (Review) |
[39] Pathological function of LOX-1 - The activation of the Rho/Rho-kinase signaling pathway plays a crucial role in regulating blood pressure and vasoconstriction, with significant implications in cardiovascular diseases such as hypertension and atherosclerosis.
|
Endothelial dysfunction
|
Rac1 (a member of the GTPase family of the Rho family) |
Not applicable (Review) |
[40] The pathological function of LOX-1 - The main role of Rac1 as a regulator of multiple cellular signalling pathways affecting cytoskeleton organization, transcription, and cell proliferation, which have implications in endothelial dysfunction and associated pathological conditions such as tumorigenesis, neurodegenerative disorders, liver cirrhosis, and cardiovascular remodelling/hypertension, has been investigated. |
LOX-1 and Nod-like receptor nucleotide-binding domain leucine rich repeat containing protein 3 (NLRP3).
|
A cellular model of Endothelial progenitor cel (EPCs) treated with different concentrations of astragaloside IV (ASV) and ox-LDL as a stimulus for cellular dysfunction. |
[57] Proinflammatory inductors - Astragaloside IV (ASV) demonstrates protective effects against oxidized low-density lipoprotein (ox-LDL)-induced dysfunction in endothelial progenitor cells (EPCs) through modulation of the LOX-1/NLRP3 pathway, highlighting its potential therapeutic role in diabetic vascular complications |
|
LOX-1 and Nod-like receptor nucleotide-binding domain leucine-rich repeat-containing protein 3 (NLRP3).
|
This is a cellular model of Endothelial progenitor cells (EPCs) treated with different concentrations of astragaloside IV (ASV) and ox-LDL as a stimulus for cellular dysfunction. |
[57] Proinflammatory inductors - Astragaloside IV (ASV) demonstrates protective effects against oxidized low-density lipoprotein (ox-LDL)-induced dysfunction in endothelial progenitor cells (EPCs) through modulation of the LOX-1/NLRP3 pathway, highlighting its potential therapeutic role in diabetic vascular complications |
|
Hypertension |
LOX-1 |
Spontaneous Hypertensive rats (SHR-SP), ---Wistar Kyoto Rats (WKY), Dahl salt-sensitive (DS) and salt-resistant rats (DR). |
[42] The pathological function of LOX-1- The increased expression of the endothelial receptor for oxidized LDL (LOX-1) in hypertensive rats suggests the involvement of LOX-1 in endothelium-dependent vasodilation dysfunction. |
No specific |
Spontaneously Hypertensive Rat (SHR) and it is Normotensive Wistar Kyoto (WKY) control |
[81] Proinflammatory inductors- The research on LOX-1 expression in spontaneously hypertensive rats (SHR) offers an invaluable opportunity to better understand the relationship between hypertension and endothelial dysfunction, opening new perspectives in cardiovascular disease pathogenesis research |
|
LOX-1 |
A molecular-cellular interaction model was reviewed to study the relationship between LDL oxidation ox-LDL, LOX-1, angiotensin II (Ang II), and Ang II receptor type 1 (AT1R) in the pathogenesis of hypertension. |
[51] Proinflammatory inductors—Hypertension is associated with vascular oxidative stress, which promotes proliferation and hypertrophy of vascular smooth muscle cells, collagen deposition, endothelial dysfunction, altered vasoconstriction, increased levels of ox-LDL, and changes in the renin-angiotensin system and angiotensin II levels. |
|
Aortic aneurysm (AA).
|
Peptide p210 related to apoB-100.
|
Apolipoprotein E-deficient (apoE-/-) mice were treated with a vaccine utilizing peptide p210 and then implanted with a pump releasing angiotensin II (Ang II) to induce AA |
[65] Expression of lox-1 and atherosclerosis - The role of apoB-containing lipoproteins in atherosclerosis was studied and suggests a potential connection with protection against abdominal aortic aneurysm through an apoB-related vaccine in mice. |
Not specified |
LOX-1
|
Atherosclerosis-susceptible C57BL/6 and atherosclerosis-resistant C3H/HeJ mice fed an atherogenic high-fat diet. (Murine macrosialin) MS does not function as an ox-LDL receptor. |
[67] Expression of lox-1 and atherosclerosis - The study shows that MS does not act as a receptor for ox-LDL on the cell surface, even though its expression increases in response to a high-fat diet and oxLDL treatment |
Chronic Venous Disease (CVD) . |
No specific
|
An observational study was conducted on 902 women from the general population, aged 45 to 54 years Evaluation of cardiovascular risk factors (ankle/brachial systolic blood pressure index (ABI) and carotid intima-media thickness of the common carotid arteries measured by ultrasound). |
[77] Relationship with atherosclerosis and thrombosis- Soluble LOX-1 levels are elevated in acute myocardial infarction, suggesting a role for the more electronegative fraction (L5) of LDL, which is associated with promoting atherogenesis and thrombus formation |
Coronary Thrombosis
|
LOX-1
|
This is an observational study that analyzed aspirated coronary thrombi from patients with STEMI and without ST-segment elevation myocardial infarction (NSTEMI) to evaluate the expression of LECTIN-LIKE OX-LDL RECEPTOR-1 (LOX-1) and the relationship between sLECTIN-LIKE OX-LDL RECEPTOR-1 (LOX-1) and L5 levels in peripheral blood.
|
[78] Relationship with atherosclerosis and thrombosis - sLOX-1 levels are elevated in acute myocardial infarction (AMI), particularly in ST-segment-elevation myocardial infarction (STEMI), suggesting a potential role for L5 in triggering atherogenesis and promoting thrombi formation |
Hypercholesterolemia.
|
LOX-1
|
Wistar rats were divided into groups, some were fed a high-cholesterol diet, and others were treated with Aegeline/atorvastatin along with the same diet. |
[63]Dyslipidemia as LOX-1 inducer- Aegeline (AG) is shown to be an effective anti-hypercholesterolemic agent by reducing Ox-LDL and LOX-1 levels, suggesting its potential as a novel diagnostic strategy for hypercholesterolemia and vascular diseases. |
And the text:
Line 502-521
The above table highlights significant aspects concerning the role of the LOX-1 receptor in various pathologies, especially in atherosclerosis and cardiovascular diseases. The expression and function of this receptor have been documented in various cell types and animal models. On the other hand, the oxidation of low-density lipoproteins (LDL) emerges as a critical factor in the development and progression of atherosclerosis, being recognized as the main ligand of LOX-1, which triggers a cascade of inflammatory effects and endothelial dysfunction. It is essential to highlight that the pathologies contemplated in the table have a direct and/or indirect interconnection with other medical conditions, such as hypertension, diabetes, and dyslipidemias, which act synergistically to enhance the development of cardiovascular diseases. In addition, studies in animal models, such as genetically modified mice and rabbits, have been used to explore the mechanisms underlying atherosclerosis and evaluate potential therapeutic interventions that could have clinical applications. Reference is also made to other target molecules, such as CD36 and SR-A, which participate with LOX-1 in lipid metabolism and the progression of these pathologies. An additional relevant implication is the close relationship between inflammation and the immune response, which plays a crucial role in the development and progression of atherosclerosis through activating the NLRP3 inflammasome and expressing pro-inflammatory cytokines. The studies collected and synthesized in this table suggest that these pathologies involve a complex interplay between genetic, environmental, and metabolic factors, endothelial dysfunction, and chronic low-grade inflammation.
Specific comments
There are many acronyms. Please verify if all these acronyms are necessary and are well used. For example, CVD and DM2 are used only two times.
R// We removed the unnecessary acronyms.
There are some perspectives as biomarker for LOX-1? Please discuss
R//We added some discussion in lines 732-857
The potential use of soluble sLOX-1 as a biomarker for acute myocardial infarction (AMI) and stroke presents a promising opportunity to enhance the current diagnostic landscape, which heavily relies on established cardiac biomarkers such as cardiac troponins and creatine kinase-MB (CK-MB).
In subjects who have suffered myocardial injury as a result of coronary angioplasty or percutaneous coronary intervention, cardiac trauma can occur during or as a consequence of the procedure, resulting in a periprocedural myocardial injury (IMPP-PCI). This injury is associated with myocardial necrosis. In a study with 214 patients undergoing percutaneous coronary intervention (PCI), it was found that 33 (15.4%) developed IMPP-PCI. Patients with IMPP-PCI had higher levels of sLOX-1 than those without IMPP-PCI (167 ± 89 vs. 99 ± 68 pg/mL; p < 0.001). Additionally, significant correlations were found between sLOX-1 levels and troponin T, CK, and CK-MB values (r = 0.677, r = 0.682; p < 0.001). These findings suggest that sLOX-1 may be an early biomarker of myocardial necrosis, and its ability to distinguish patients with IMPP-PCI enhances its value as a diagnostic tool in risk stratification and the development of personalized therapeutic strategies. (Balin et al., 2012)
In conditions such as myocardial necrosis, especially in the early stage of acute myocardial infarction with ST-segment elevation (STEMI), the sensitivity of some important markers such as creatine kinase MB (CK-MB), cardiac troponins, myoglobin, and heart-type fatty acid-binding protein (H-FABP) has been analyzed. In the study by Nobuaki Kobayashi and colleagues, plasma levels of sLOX-1 were evaluated in 125 patients with STEMI, 44 with non-ST-segment elevation myocardial infarction (NSTEMI), and 125 without acute myocardial infarction (non-AMI). It was found that sLOX-1 levels were significantly higher in patients with STEMI and NSTEMI compared to those without AMI (median, 25th, and 75th percentiles: 241.0, 132.3, and 472.2 vs. 147.3, 92.9, and 262.4 vs. 64.3, 54.4 and 84.3 pg/mL, respectively). These results highlight the discriminative capacity of sLOX-1 between STEMI and non-AMI, with a sensitivity of 89.6% and a specificity of 82.4%, using an optimal cutoff value of 91.0 pg/mL. In conclusion, the findings of this study suggest that sLOX-1 may be a highly sensitive and specific biomarker for the early diagnosis of STEMI, surpassing other traditional biomarkers.(Kobayashi et al., 2011)
The most important findings are from Angela Pirillo and colleagues. The review was: sLOX-1 showed higher sensitivity and specificity than cardiac troponin T (TnT) and Heart-type Fatty Acid Binding Protein (H-FABP) and could detect Acute Coronary Syndrome (ACS) in subjects with non-significantly elevated TnT values. In ACS patients, H-FABP values correlated significantly with TnT values. In contrast, sLOX-1 values did not correlate with TnT or H-FABP, indicating its expression is independent of the mentioned traditional markers. Additionally, plasma levels of sLOX-1 were significantly higher in subjects with acute myocardial infarction (AMI) than those without AMI. They were significantly higher in subjects with ST-segment elevation myocardial infarction (STEMI) than in the non-ST-segment elevation myocardial infarction (NSTEMI) group. In AMI patients, sLOX-1 levels increased in the early stages of acute myocardial infarction, persisted for 24 hours after arrival at the emergency room, and declined to baseline levels 16 days after the onset of STEMI. sLOX-1 levels were elevated in patients with acute aortic dissection (AAD) and in patients with acute coronary syndrome without ST-segment elevation (NSTEACS) compared to controls. In detecting subjects with AAD and distinguishing them from those with NSTEACS, sLOX-1 proved to be a better predictor than TnT. Therefore, sLOX-1 is suggested as a sensitive and specific biomarker in diagnosing acute coronary syndromes and acute aortic dissection. It has important implications for accelerating their identification and preventing potentially fatal complications in subjects with atherosclerosis-related conditions.(Pirillo & Catapano, 2013)
In another study, sLOX-1 levels were analyzed in overweight/obese children and adolescents, finding that these levels were elevated during and after puberty compared to the general population. Additionally, sLOX-1 was positively associated with inflammatory markers and unfavorable cardiometabolic risk profiles, such as insulin resistance, dyslipidemia, and hypertension. These findings suggest that sLOX-1 could play an important role as an early biomarker of cardiometabolic risk and inflammation in overweight/obese children and adolescents (Stinson et al., 2023)
On the other hand, patients with acute coronary syndromes (ACS) were examined to assess the role of vascular inflammation and biomarkers, specifically sLOX-1, in the development and complexity of coronary artery disease (CAD). Data were collected from patients admitted to the emergency department with unstable angina or NSTE-ACS. Elevated levels of high-sensitivity C-reactive protein (hs-CRP) and sLOX-1 were associated with more complex CAD, as per the modified Gensini score; this suggests that vascular inflammation, indicated by sLOX-1, could play a crucial role in risk prediction and ACS management, highlighting its significance as a novel predictive biomarker in individuals with this disease.(Çoner et al., 2020)
Incremental Value of sLOX-1 Over Established Biomarkers
- Early Detection: Cardiac troponins are the gold standard for diagnosing AMI due to their high sensitivity and specificity for myocardial injury. However, troponins may only elevate several hours after the onset of AMI. sLOX-1, by contrast, might rise earlier in the course of endothelial activation and oxidative stress, potentially allowing for quicker diagnosis and intervention.
- Sensitivity to Subclinical Atherosclerosis: Unlike troponins and CK-MB, which increase in response to myocardial damage, sLOX-1 levels reflect endothelial dysfunction and oxidative stress, which are early events in atherosclerosis; this makes sLOX-1 a potentially valuable biomarker for identifying subclinical atherosclerosis and predicting cardiovascular events, even before structural heart damage occurs.
- Prognostic Value: Studies have suggested that elevated levels of sLOX-1 are associated with an increased risk of future cardiovascular events, providing prognostic information beyond the acute setting; this could be particularly useful for risk stratification and long-term management of patients with coronary artery disease.
Integrating sLOX-1 into Clinical Practice
- Diagnostic Algorithms: Integrating sLOX-1 testing into routine diagnostic algorithms for AMI and stroke could complement the use of current biomarkers, potentially leading to earlier and more accurate risk stratification and therapeutic interventions. For example, an elevated sLOX-1 level in a patient with borderline troponin levels could prompt more aggressive management or further diagnostic testing.
- Feasibility: Integrating sLOX-1 testing is feasible only if rapid, cost-effective, and reliable assays are developed. Current enzyme-linked immunosorbent assay (ELISA) kits for sLOX-1 are primarily used in research settings and must be adapted for routine clinical use.
Challenges and Opportunities for Implementing LOX-1 as a Routine Biomarker
- Standardization: A significant challenge is the lack of standardized thresholds for sLOX-1 levels across different populations and clinical settings. Establishing these norms would require extensive validation studies.
- Cost and Accessibility: As with any new diagnostic tool, cost-effectiveness studies would be crucial to justify the inclusion of sLOX-1 testing in standardized diagnostic protocols, particularly in resource-limited settings.
- Education and Awareness: To ensure its effective use, clinicians must be educated about the implications of sLOX-1 levels and their integration into clinical decision-making processes.
- Regulatory Approval: Gaining regulatory approval for clinical use involves rigorous testing and validation to meet safety and efficacy standards, which can be time-consuming and costly.
Future Prospects
The potential for sLOX-1 to become a routine biomarker shortly will largely depend on ongoing and future clinical trials that can definitively demonstrate its added value in clinical settings. With positive results, sLOX-1 could fill a crucial gap in the early diagnosis and risk stratification of cardiovascular diseases, ultimately leading to better patient outcomes.
In conclusion, it is essential to highlight the role of sLOX-1 measurements in clinical practice as they offer a window into a more precise evaluation of medical conditions related to the development and progression of diseases. Considering its potential, as found in the literature, it is highly feasible to integrate sLOX-1 testing into routine diagnostic algorithms for acute myocardial infarction (AMI) and stroke. The introduction of sLOX-1 into these algorithms could present unique challenges and opportunities. On the one hand, we may face obstacles such as additional validation of its effectiveness in different clinical contexts and standardization of testing procedures. However, the successful integration of sLOX-1 could open new doors to earlier and more accurate detection of AMI and stroke, potentially improving clinical outcomes and reducing the burden on healthcare systems. Achieving this goal will require interdisciplinary collaboration, ongoing research, and careful assessment of associated benefits and risks. According to reported findings, significant advances in incorporating sLOX-1 as a routine biomarker could be seen shortly, marking a step forward in personalized and precision healthcare with simple tests such as ELISA in patient plasma.

Reviewer 2 Report
Comments and Suggestions for Authors
Reviewer comments and suggestions
The authors in this study explained that the LOX-1 which is the main receptor for oxidized low-density lipoproteins (oxLDL) and its involvement with cardiovascular diseases. LOX-1 belongs to the Scavenger receptors (SR), which are associated with various cardiovascular pathologies; the most important and severe is the atherosclerotic plaque formation in the intimal layer of the endothelium, evolving into a complicated thrombus that may cause total obstruction in the lumen of the blood vessels, leading to oxygen deprivation to the heart. The development of atherosclerosis has been the most relevant due to its relationship with cerebrovascular accidents and heart attacks. In this review, the authors summarized the findings connected with the physiologic and pathophysiological processes of LOX-1 to support the detection, diagnosis, and prevention of those pathologies.
Overall, the manuscript needs to be well written, as in this form it cannot be accepted. I have mentioned a few major concerns/comments that need to be explained or modified
- Lines 3-5 Title should be modified based on the content
- Line 22-28 Please avoid long sentences in the manuscript also line number 41-48
- Line 33 Please write up in the best possible way so that you may cover the manuscript "those pathologies" is not the best word choice.
- Line 64 Please combine the reference based on MDPI
- Lines 307 to 321 also need a proper reference to cite
- Line 326-329 The authors can add up a few references from where the figure idea was taken or the newly drawn
- Line 355-356 which study the authors were talking, it should be well written for the common reader along with the useful citation
- Please check lines 373-385 do not consist of any references
- The authors can enhance the manuscript by adding more recent references as it’s a review article.
- Please check the representation of NF-KB, LINE 263, the name of this transcriptional factor should be the same at all places.
- All references should be modified based on the MDPI guidelines
Author Response
We thank the reviewer for the excellent comments on our manuscript. Below we describe how we have included the comments and references:
- Lines 3-5 Title should be modified based on the content :
R// We suggest the title:
“LOX-1 in Cardiovascular Disease: A Comprehensive Molecular and Clinical Review"
- Line 22-28 Please avoid long sentences in the manuscript also line number 41-48
R// Line 22-28:
LOX-1 belongs to the Scavenger receptors (SR), which are associated with various cardiovascular pathologies; the most important and severe is the atherosclerotic plaque formation in the intimal layer of the endothelium, evolving into a complicated thrombus with the participation of fibroblasts, activated platelets, apoptotic muscle cells and macrophages transformed into foam cells, causing changes in the vascular endothelium homeostasis, producing partial or total obstruction in the lumen of the blood vessels, leading to oxygen deprivation to the heart
Changed to;
LOX-1 belongs to the scavenger receptors (SR), which are associated with various cardiovascular pathologies. The most important and severe of these is the formation of atherosclerotic plaques in the intimal layer of the endothelium. These plaques can evolve into complicated thrombi with the participation of fibroblasts, activated platelets, apoptotic muscle cells, and macrophages transformed into foam cells. This process causes changes in vascular endothelial homeostasis, leading to partial or total obstruction in the lumen of blood vessels. This obstruction can result in oxygen deprivation to the heart.
Line 41-48
LOX-1, ORL-1, or lectin-like oxidized low-density lipoprotein receptor one is the main receptor for oxidized low-density lipoproteins (ox-LDL) circulating in the bloodstream that come from food intake. It belongs to the scavenger receptors (SR) family [1], which are associated with various cardiovascular pathologies; the most important and severe is the formation of atherosclerotic plaque in the intima layer of the endothelium, evolving into a complicated thrombus with the participation of fibroblasts, activated platelets, apoptotic muscle cells, and macrophages transformed into foam cells, causing changes in the vascular endothelium homeostasis, producing partial or total obstruction in the lumen of the blood vessels, that is, the reduction of blood flow to the heart muscle, producing myocardial ischemia.
Changed to :
LOX-1, also known as ORL-1 or lectin-like oxidized low-density lipoprotein receptor one, is the primary receptor for oxidized low-density lipoproteins (oxLDL) circulating in the bloodstream, originating from food intake. It belongs to the scavenger receptors (SR) family [1], associated with various cardiovascular pathologies. The most important and severe of these is the formation of atherosclerotic plaques in the intimal layer of the endothelium, which can evolve into complicated thrombi with the participation of fibroblasts, activated platelets, apoptotic muscle cells, and macrophages transformed into foam cells. This process causes changes in vascular endothelial homeostasis, leading to partial or total obstruction in the lumen of blood vessels, resulting in reduced blood flow to the heart muscle and myocardial ischemia.
- Line 33 Please write up in the best possible way so that you may cover the manuscript "those pathologies" is not the best word choice.
R//We change “those pathologies” for “those diseases”
- Line 64 Please combine the reference based on MDPI
R// whe have combined the references: [7] [8] [9] change to [7- 9]
- Lines 307 to 321 also need a proper reference to cite
- Line 434: Y. Lu et al., ‘The Functional Role of Lipoproteins in Atherosclerosis: Novel Directions for Diagnosis and Targeting Therapy’, Aging Dis, vol. 13, no. 2, p. 491, 2022, doi: 10.14336/AD.2021.0929.
- Roy, M. Orecchioni, and K. Ley, ‘How the immune system shapes atherosclerosis: roles of innate and adap-tive immunity’, Nat Rev Immunol, vol. 22, no. 4, pp. 251–265, Apr. 2022, doi: 10.1038/s41577-021-00584-1.
- Hasheminasabgorji and J. C. Jha, ‘Dyslipidemia, diabetes and atherosclerosis: Role of inflammation and ros-redox-sensitive factors’, Biomedicines, vol. 9, no. 11. 2021. doi: 10.3390/biomedicines9111602.
- Line 326-329 The authors can add up a few references from where the figure idea was taken or the newly drawn(
R//Adapted from “Atherosclerosis Progression,” by BioRender.com (2023). Retrieved from https://app.biorender.com/biorender-templates
- Line 355-356 which study the authors were talking, it should be well written for the common reader along with the useful citation
R//
the increasing expression of the low-density lipoprotein receptor similar to lectin LOX-1 in knockout mice for Apo-E fed a high-cholesterol diet, showing extensive atherosclerosis determined by Sudan IV staining. In the other group, the reduction of atherosclerosis was up to 67% with antihypercholesterolemic drugs, demonstrating the relationship between the LOX-1 expression and p38 MAPK activation.
The text changed to
“J. Chen et al. found that in genetically modified mice fed a high-cholesterol diet, there was an increase in the expression of the low-density lipoprotein receptor similar to lectin LOX-1 in the arteries, leading to extensive atherosclerosis, while, in another group of mice with similar atherosclerosis but treated with antihypercholesterolemic drugs, there was a reduction in atherosclerosis of up to 67%. This demonstrates a relationship between the expression of LOX-1 and the activation of p38 MAPK in atherosclerosis”.
- Please check lines 373-385 do not consist of any references
R//We added the reference:
- Malekmohammad, E. E. Bezsonov, and M. Rafieian-Kopaei, ‘Role of Lipid Accumulation and Inflammation in Atherosclerosis: Focus on Molecular and Cellular Mechanisms’, Front Cardiovasc Med, vol. 8, Sep. 2021, doi: 10.3389/fcvm.2021.707529.
- The authors can enhance the manuscript by adding more recent references as it’s a review article.
We added the following references:
[1] S. Balzan and V. Lubrano, “LOX-1 receptor: A potential link in atherosclerosis and cancer,” Life Sci, vol. 198, pp. 79–86, Apr. 2018, doi: 10.1016/j.lfs.2018.02.024.
[2] T. Kita, “LOX-1, a possible clue to the missing link between hypertension and atherogenesis,” Circ Res, vol. 84, no. 9, pp. 1113–1115, 1999, doi: 10.1161/01.RES.84.9.1113.
[3] M. Yan, J. L. Mehta, W. Zhang, and C. Hu, “LOX-1, Oxidative Stress and Inflammation: A Novel Mechanism for Diabetic Cardiovascular Complications,” Cardiovasc Drugs Ther, vol. 25, no. 5, pp. 451–459, Oct. 2011, doi: 10.1007/s10557-011-6342-4.
[4] A. L. Vavere et al., “Lectin‐Like Oxidized Low‐Density Lipoprotein Receptor 1 Inhibition in Type 2 Diabetes: Phase 1 Results,” J Am Heart Assoc, vol. 12, no. 3, Feb. 2023, doi: 10.1161/JAHA.122.027540.
[5] M. Miyawaki et al., “‘Benifuuki’ Extract Reduces Serum Levels of Lectin-Like Oxidized Low-Density Lipoprotein Receptor-1 Ligands Containing Apolipoprotein B: A Double-Blind Placebo-Controlled Randomized Trial,” Nutrients, vol. 10, no. 7, p. 924, Jul. 2018, doi: 10.3390/nu10070924.
[6] K. O. Murray, K. R. Ludwig, S. Darvish, M. E. Coppock, D. R. Seals, and M. J. Rossman, “Chronic mitochondria antioxidant treatment in older adults alters the circulating milieu to improve endothelial cell function and mitochondrial oxidative stress,” American Journal of Physiology-Heart and Circulatory Physiology, vol. 325, no. 1, pp. H187–H194, Jul. 2023, doi: 10.1152/ajpheart.00270.2023.
[7] X. Wang, M. I. Phillips, and J. L. Mehta, “LOX-1 and Angiotensin Receptors, and Their Interplay,” Cardiovasc Drugs Ther, vol. 25, no. 5, p. 401, Oct. 2011, doi: 10.1007/s10557-011-6331-7.
[8] T. KITA et al., “Oxidized‐LDL and Atherosclerosis: Role of LOX‐1,” Ann N Y Acad Sci, vol. 902, no. 1, pp. 95–102, May 2000, doi: 10.1111/j.1749-6632.2000.tb06304.x.
[9] A. J. Berberich and R. A. Hegele, “A Modern Approach to Dyslipidemia,” Endocr Rev, vol. 43, no. 4, pp. 611–653, Jul. 2022, doi: 10.1210/endrev/bnab037.
[10] J. S. Dron and R. A. Hegele, “Polygenic influences on dyslipidemias,” Curr Opin Lipidol, vol. 29, no. 2, pp. 133–143, Apr. 2018, doi: 10.1097/MOL.0000000000000482.
[11] Z. Hoyk et al., “Cerebrovascular Pathology in Hypertriglyceridemic APOB-100 Transgenic Mice,” Front Cell Neurosci, vol. 12, Oct. 2018, doi: 10.3389/fncel.2018.00380.
[12] A. Hirata et al., “The relationship between serum levels of LOX-1 ligand containing ApoAI as a novel marker of dysfunctional HDL and coronary artery calcification in middle-aged Japanese men,” Atherosclerosis, vol. 313, pp. 20–25, Nov. 2020, doi: 10.1016/j.atherosclerosis.2020.09.013.
- Please check the representation of NF-KB, LINE 263. The name of this transcriptional factor should be the same everywhere.
We eliminated (the nuclear factor enhancer of kappa light chains of activated B cells) in lines 416-417. and the name is the same in all places.
- All references should be modified based on the MDPI guidelines
The references were revised and modified based on the MDPI guidelines

Reviewer 3 Report
Comments and Suggestions for Authors
Dear Authors,
This is a very well-written and interesting review. It is rare that a reviewer does not have many comments, but this work appears to be complete after some minor modifications
The article is clear, relevant, and presented in a well-structured manner. Although there are some adjustments that I would like to suggest.
1. In 2.1 emphasize how (if) specific mutations or isoforms of LOX-1 contribute to disease pathogenesis or progression.
2. In 3. and 4.1.1. Expand on the clinical implications of LOX-1 function in atherosclerosis and related diseases. Discuss potential therapeutic targets or interventions based on the modulation of scavenger receptor activity, particularly in the context of cardiovascular diseases. Expand on therapeutic implications of targeting LOX-1 in inflammatory diseases, discussing potential interventions (e.g., pharmacological inhibitors, gene therapy) aimed at modulating LOX-1 expression or activity to attenuate disease progression. Incorporate references to clinical trials or translational studies exploring LOX-1-targeted therapies in human subjects, demonstrating the translational potential of the discussed molecular insights into clinical practice.
3. In 4.2. Provide clearer explanations of the molecular mechanisms underlying dyslipidemia-induced LOX-1 expression. Elucidate how dyslipidemia-related molecules (e.g., oxidized LDL, lysophosphatidylcholine) promote LOX-1 expression and contribute to atherosclerosis progression. Discuss potential pharmacological interventions or lifestyle modifications aimed at modulating LOX-1 expression to reduce cardiovascular risk. Discuss monogenic vs. polygenic dyslipidemias and their implications for personalized medicine.
4. In 4.3.1: Clarify the interactions between LOX-1 and specific lipoproteins (e.g., apoB100, apoAI) implicated in foam cell formation and plaque development. Consider discussing emerging therapeutic strategies aimed at mitigating LOX-1-mediated atherosclerosis and thrombotic events.
5. In 5. Include (if available) a comparative analysis with established cardiac biomarkers (e.g., cardiac troponins, CK-MB) to demonstrate the incremental value of sLOX-1 in clinical practice. Emphasize the translation of sLOX-1 measurements into clinical practice, discussing the feasibility of integrating sLOX-1 testing into routine diagnostic algorithms for AMI and stroke. What could be the potential challenges and opportunities for implementing LOX-1 as a routine biomarker. Can this be done in the near future?
Reviewer
Comments on the Quality of English Language
Please read the article carefully, as there are minor errors and typos.
Author Response
We thank the reviewer for the excellent comments on our manuscript. Below we describe how we have included the comments and references:
This is a very well-written and interesting review. It is rare that a reviewer does not have many comments, but this work appears to be complete after some minor modifications
The article is clear, relevant, and presented in a well-structured manner. Although there are some adjustments that I would like to suggest.
- In 2.1 emphasize how (if) specific mutations or isoforms of LOX-1 contribute to disease pathogenesis or progression.
We included the following discussion:
Lines: 174-213
How do specific mutations or isoforms of LOX-1 contribute to disease pathogenesis or progression?
The LOX-1 receptor is crucial in developing atherosclerosis and related cardiovascular conditions. Mutations and alternative splicing resulting in different isoforms of LOX-1 can significantly influence the receptor's function, thereby affecting disease pathogen-esis and progression.
Mutations of LOX-1
- Impact on LOX-1 Functionality: Mutations in the OLR1 gene can alter the receptor's structure and function. For example, specific mutations may affect the receptor's ability to bind to ox-LDL, either enhancing or inhibiting this critical interaction. Enhanced binding capacity can lead to increased formation of foam cells, accelerating atherosclerotic plaque development. Conversely, mutations that reduce LOX-1's affinity-ty for ox-LDL could confer protective effects against atherosclerosis.
- Clinical Relevance of Mutations: Specific mutations in the OLR1 gene have been associated with an increased risk of cardiovascular diseases. For instance, specific polymorphisms linked to the OLR-1 gene have been correlated with higher susceptibility to coronary artery disease and myocardial infarction. Understanding these genetic variations can help identify individuals at higher risk and lead to targeted prevention strategies.
Isoforms of LOX-1
- LOXIN Isoform: An important alternative splicing event within the OLR1 gene leads to the production of an isoform known as LOXIN. This isoform lacks a portion of the C-terminal domain, which is crucial for the receptor's ability to bind ox-LDL. LOX-IN acts as a dominant-negative regulator of the full-length LOX-1 receptor, potentially reducing ox-LDL uptake and subsequent foam cell formation. This suggests that higher levels of LOXIN relative to the full-length LOX-1 could be protective against atherosclerosis.
- Therapeutic Implications of LOXIN: Enhancing the expression of LOXIN or administering it as a therapeutic protein could provide a novel approach to preventing or treating atherosclerosis. By competing with the full-length LOX-1 for binding sites, LOXIN could reduce ox-LDL uptake and the ensuing inflammatory responses in the vascular endothelium.
In conclusion, the research on mutations and isoforms of LOX-1 reveals their importance in cardiovascular diseases. These variations offer a deeper understanding of these diseases' pathogenesis and suggest new therapeutic opportunities. Strategies that modulate the expression levels of LOXIN or mimic its effects could present innovative approaches for preventing and treating atherosclerosis and related conditions. Studying these mutations and isoforms provides a solid foundation for more personalized and effective therapeutic interventions in cardiovascular diseases, opening up new avenues for improving overall cardiovascular health.
- In 3. and 4.1.1. Expand on the clinical implications of LOX-1 function in atherosclerosis and related diseases. Discuss potential therapeutic targets or interventions based on the modulation of scavenger receptor activity, particularly in the context of cardiovascular diseases. Expand on therapeutic implications of targeting LOX-1 in inflammatory diseases, discussing potential interventions (e.g., pharmacological inhibitors, gene therapy) aimed at modulating LOX-1 expression or activity to attenuate disease progression. Incorporate references to clinical trials or translational studies exploring LOX-1-targeted therapies in human subjects, demonstrating the translational potential of the discussed molecular insights into clinical practice.
We added the following discussion:
Line 343-417
LOX-1 function in atherosclerosis and related diseases: Excessive ROS production con-tributes to atherosclerosis and, in the presence of lipid peroxidation and DNA damage, is a factor in cancer development. Activation of LOX-1 stimulates the expression of proinflammatory and proangiogenic molecules, such as NF-kB and VEGF, in endothe-lial cells and macrophages. LOX-1 is implicated in cardiovascular diseases and various types of cancer. Inhibition of LOX-1 has been shown to reduce NF-kB activation and inflammatory pathways, suggesting a link between atherosclerosis and cell transfor-mation. Additionally, remodeling proteins like MMP-2 and MMP-9 increase angiogen-esis in atherosclerotic plaque and human prostate cancer cells. Inhibiting LOX-1 ex-pression could represent a promising strategy for treating atherosclerosis and cancer [56].
In hypertension, it has been demonstrated that hypertensive patients are more suscep-tible to LDL oxidation, have high levels of Ang II, and have shown a higher incidence of myocardial infarction, while normotensive controls have a lower susceptibility to LDL oxidation. A disadvantage in hypertensive patients is that, due to high levels of Ang II, they have a greater direct induction of oxidative stress and inflammatory re-sponse in the aorta, thus increasing the uptake of ox-LDL by endothelial cells through positive regulation of LOX-1, leading to endothelial damage, which are responsible for maintaining vascular tone and whose molecules involved in such events are prostacy-clin (PGI2), endothelin (ET), Ang II, and NO. Vasomotor tone regulation appears to be controlled by the constant action of NO. Inhibition of NO and PGI2 formation allows opposing forces of vasodilation, such as ET, Ang II, or thromboxane A2, to determine the artery's ability to maintain its lumen in the presence of changing forces caused by the formation and progression of atherosclerosis lesions [57].
Diabetes mellitus is a common metabolic disease characterized by a state of oxidative stress, inflammation, and endothelial dysfunction, which can lead to various complica-tions such as ischemic heart disease, nephropathy, neuropathy, retinopathy, and im-paired wound healing. LOX is critical in multiple signal transduction pathways and is involved in oxidative stress and inflammation. The findings reveal the role of LOX-1 in the development and progression of diabetic vasculopathy, which is the underlying mechanism of diabetic complications [58].
The overproduction of reactive oxygen species (ROS) contributes to atherosclerosis and cancer by damaging cells and promoting inflammation. The LOX-1 receptor plays a significant role in this process by stimulating inflammation and the formation of new blood vessels. In diseases such as hypertension and diabetes, LOX-1 is involved in vas-cular dysfunction and the progression of cardiovascular complications. Blocking LOX-1 could be a promising strategy to treat these diseases and reduce the risk of complica-tions.
Some potential therapeutic targets or interventions could be:
Creation of specific pharmacological LOX-1 inhibitors: The development of drugs fo-cused on blocking the activity of LOX-1 would reduce disease inflammation in condi-tions such as atherosclerosis and cancer.
- Application of Gene Therapy: Genetic modification to regulate the expression of LOX-1 could be a strategy to control its function in the body and prevent its negative effects on cardiovascular diseases and cancer.
For example, in a phase 1 study, MEDI6570, a monoclonal antibody targeting LOX-1, was evaluated for safety and efficacy in patients with type 2 diabetes and atherosclerosis. Dose-dependent reductions in soluble LOX-1 levels were observed, with a favourable safety profile and nonlinear pharmacokinetics. MEDI6570 showed promising results in suppressing soluble LOX-1 levels and demonstrated a pharmacokinetic profile consistent with once-monthly dosing, suggesting its potential as a therapeutic option [59].
- Diet and lifestyle modulation: Starting a healthy lifestyle and a diet rich in antioxidants could help reduce LOX-1 activation and its impact on cardiovascular health and cancer risk.
For example, a study investigated the effects of "Benifuuki" green tea extract, rich in O-methylated catechins, on cholesterol levels in healthy volunteers. Participants were divided into groups consuming high-dose extract, low-dose extract, or no extract for 12 weeks. Results showed a significant reduction in lectin-like oxidized LDL receptor-1 ligand-containing ApoB (LAB) levels in the high-dose group compared to the other groups. However, there were no significant differences in total cholesterol, triglycerides, and LDL cholesterol levels between the groups. The findings suggest that "Benifuuki" extract may help prevent arteriosclerosis by reducing LAB levels [60].
- Consumption of anti-inflammatory agents: These agents could have beneficial effects by reducing the activation of LOX-1 and its contribution to diseases such as atherosclerosis and cancer.
For example, a study investigated the impact of the mitochondria-targeted antioxidant (MitoQ) on endothelial function in older adults. It was found that MitoQ treatment reduced levels of mitochondrial reactive oxygen species (mtROS), improving endothelial function, and lowering circulating levels of ox-LDL. The analysis showed that exposure of endothelial cells to plasma from individuals treated with MitoQ resulted in increased nitric oxide (NO) production and decreased mtROS activity compared to placebo. These findings suggest that MitoQ enhances endothelial function by reducing oxLDL and improving NO production. This study provides a deeper understanding of the mechanisms underlying the improvement of age-related endothelial dysfunction through MitoQ treatment [61].
These interventions have the potential to modulate LOX-1 activity and could be explored as possible therapeutic strategies in the future to treat various diseases related to this receptor.
- In 4.2. Provide clearer explanations of the molecular mechanisms underlying dyslipidemia-induced LOX-1 expression. Elucidate how dyslipidemia-related molecules (e.g., oxidized LDL, lysophosphatidylcholine) promote LOX-1 expression and contribute to atherosclerosis progression. Discuss potential pharmacological interventions or lifestyle modifications aimed at modulating LOX-1 expression to reduce cardiovascular risk. Discuss monogenic vs. polygenic dyslipidemias and their implications for personalized medicine.
R// We added the following discussion:
Line 487-533
The dyslipidemia-induced expression of LOX-1 involves a series of molecular mechanisms that can be triggered by elevated levels of lipids in the bloodstream. One of the primary contributors is the presence of oxidized low-density lipoproteins and other abnormal lipids in the circulation. These abnormal lipids can directly interact with cell receptors, such as LOX-1, triggering its expression.
Various molecular mechanisms are involved in the expression of LOX-1 in dyslipidemias, with higher LDL availability coexisting with an increased risk of atherosclerosis development. The renin-angiotensin system (RAS) regulates blood pressure, water-salt balance, and cardiovascular disease pathogenesis. Angiotensin II acts as its active mediator, primarily through type 1 and type 2 receptors. AT1R activation mediates most pathophysiological effects of Ang II, while the precise function of AT2R remains unclear, generally considered opposing AT1R effects. LOX-1 binds and degrades ox-LDL, linked to endothelial dysfunction, fibroblast growth, and vascular smooth muscle cell hypertrophy, crucial in atherosclerosis, hypertension, and myocardial remodeling. Evidence suggests an interaction between LOX-1 and Ang II receptors, reciprocally regulating expression and activity. Signals such as reactive oxygen species, nitric oxide, protein kinase C, and mitogen-activated protein kinases mediate this interaction, impacting dyslipidemia and RAS activation [68].
On the other hand, the interaction between oxidized LDL and its receptor, LOX-1, plays a crucial role in early-stage atherogenesis. Lysophosphatidylcholine (lyso-PC), an increased component in atherosclerotic lipoproteins, can upregulate adhesion molecules and growth factors for monocytes and T lymphocytes. This interaction highlights the significance of LOX-1 in mediating the effects of oxidized LDL in atherosclerosis development.
In summary, dyslipidemia-induced LOX-1 expression involves the interaction of ox-LDL and other abnormal lipids with specific cell receptors, as well as the activation of intracellular signaling pathways that regulate LOX-1 gene transcription. These molecular mechanisms contribute to the development and progression of atherosclerosis in individuals with dyslipidemia [69].
It is important to emphasize the significance of recognizing monogenic and polygenic dyslipidemias in clinical practice. Monogenic dyslipidemias are rare, while common genetic variants have a significant impact on lipid traits under a polygenic framework. This suggests that a considerable proportion of the population with dyslipidemia may have a genetic predisposition contributing to their abnormal lipid profiles [70].
The role of small common genetic effects, especially Single Nucleotide Polymorphisms (SNPs), in polygenic dyslipidemias. It emphasizes that these SNPs, while individually having a modest effect, can have a significant impact when considered collectively, especially in individuals with clinical dyslipidemia. Therefore, it highlights the importance of understanding the contribution of common genetic factors to predisposition to dyslipidemias [71].
The complexity of dyslipidemias and the need for an integrated approach considering genetic and environmental factors. While advances in identifying genetic variants have improved our understanding of the genetic basis of dyslipidemias, there is still much to be understood about how these variants interact with environmental factors and contribute to cardiovascular risk. Personalized medicine can benefit from a thorough assessment of genetic profiles and an understanding of how these profiles influence treatment response and prognosis in patients with dyslipidemia.
- In 4.3.1: Clarify the interactions between LOX-1 and specific lipoproteins (e.g., apoB100, apoAI) implicated in foam cell formation and plaque development. Consider discussing emerging therapeutic strategies aimed at mitigating LOX-1-mediated atherosclerosis and thrombotic events.
R// We added the following discussion:
Line 665-680.
The interactions between LOX-1 and specific lipoproteins, such as apoB100 and apoAI, play a crucial role in foam cell formation and the development of atherosclerotic plaques. LOX-1 is primarily responsible for binding and internalizing ox-LDL, leading to foam cell formation, a hallmark of early atherogenesis. Additionally, LOX-1 can interact with other molecules, including apoB100, contributing to the uptake of modified lipoproteins by endothelial cells and macrophages within the arterial wall [92, 93].
Emerging therapeutic strategies to mitigate LOX-1-mediated atherosclerosis and thrombotic events focus on various process stages. One approach involves blocking the interaction between LOX-1 and ox-LDL using monoclonal antibodies or small molecule inhibitors. Another strategy is to modulate the expression or activity of LOX-1 through pharmacological agents or lifestyle interventions. In addition, research is directed to-ward interventions targeting downstream signaling pathways activated by LOX-1, such as those related to inflammation and oxidative stress.
Understanding the interactions between LOX-1 and specific lipoproteins provides insights into developing novel therapeutic approaches to combat atherosclerosis and reduce the risk of thrombotic events associated with this condition.
- In 5. Include (if available) a comparative analysis with established cardiac biomarkers (e.g., cardiac troponins, CK-MB) to demonstrate the incremental value of sLOX-1 in clinical practice. Emphasize the translation of sLOX-1 measurements into clinical practice, discussing the feasibility of integrating sLOX-1 testing into routine diagnostic algorithms for AMI and stroke. What could be the potential challenges and opportunities for implementing LOX-1 as a routine biomarker. Can this be done in the near future?
R// We added the following discussion:
Line 732-857:
The potential use of soluble sLOX-1 as a biomarker for acute myocardial infarction and stroke presents a promising opportunity to enhance the current diagnostic landscape, which relies on established cardiac biomarkers such as cardiac troponins and creatine kinase-MB (CK-MB).
In subjects who have suffered myocardial injury as a result of coronary angioplasty or percutaneous coronary intervention, cardiac trauma can occur during or as a consequence of the procedure, resulting in a periprocedural myocardial injury (IMPP-PCI). This injury is associated with myocardial necrosis. In a study with 214 patients undergoing percutaneous coronary intervention, it was found that 33 (15.4%) developed IMPP-PCI. Patients with IMPP-PCI had higher levels of sLOX-1 than those without IMPP-PCI (167 ± 89 vs. 99 ± 68 pg/mL; p < 0.001). Additionally, significant correlations were found between sLOX-1 levels and troponin T, CK, and CK-MB values (r = 0.677, r = 0.682; p < 0.001). These findings suggest that sLOX-1 may be an early biomarker of myocardial necrosis, and its ability to distinguish patients with IMPP-PCI enhances its value as a diagnostic tool in risk stratification and the development of personalized therapeutic strategies [97]
In conditions such as myocardial necrosis, especially in the early stage of acute myo-cardial infarction with ST-segment elevation (STEMI), the sensitivity of some im-portant markers such as creatine kinase MB, cardiac troponins, myoglobin, and heart-type fatty acid-binding protein (H-FABP) has been analyzed. In the study by Nobuaki Kobayashi and colleagues, plasma levels of sLOX-1 were evaluated in 125 patients with STEMI, 44 with non-ST-segment elevation myocardial infarction (NSTEMI), and 125 without acute myocardial infarction (non-AMI). It was found that sLOX-1 levels were significantly higher in patients with STEMI and NSTEMI compared to those without AMI (median, 25th, and 75th percentiles: 241.0, 132.3, and 472.2 vs. 147.3, 92.9, and 262.4 vs. 64.3, 54.4 and 84.3 pg/mL, respectively). These results highlight the discriminative capacity of sLOX-1 between STEMI and non-AMI, with a sensitivity of 89.6% and a specificity of 82.4%, using an optimal cutoff value of 91.0 pg/mL. In con-clusion, the findings of this study suggest that sLOX-1 may be a highly sensitive and specific biomarker for the early diagnosis of STEMI, surpassing other traditional bi-omarkers [95].
The most important findings are from Angela Pirillo and colleagues. The review was: sLOX-1 showed higher sensitivity and specificity than cardiac troponin T (TnT) and Heart-type Fatty Acid Binding Protein (H-FABP) and could detect Acute Coronary Syndrome (ACS) in subjects with non-significantly elevated TnT values. In ACS pa-tients, H-FABP values correlated significantly with TnT values. In contrast, sLOX-1 values did not correlate with TnT or H-FABP, indicating its expression is independent of the mentioned traditional markers. Additionally, plasma levels of sLOX-1 were sig-nificantly higher in subjects with acute myocardial infarction (AMI) than those with-out AMI. They were significantly higher in subjects with ST-segment elevation myo-cardial infarction than in the non-ST-segment elevation myocardial infarction group. In AMI patients, sLOX-1 levels increased in the early stages of acute myocardial infarc-tion, persisted for 24 hours after arrival at the emergency room, and declined to base-line levels 16 days after the onset of STEMI. sLOX-1 levels were elevated in patients with acute aortic dissection (AAD)and in patients with an acute coronary syndrome without ST-segment elevation (NSTEACS) compared to controls. In detecting subjects with AAD and distinguishing them from those with NSTEACS, sLOX-1 proved to be a better predictor than TnT. Therefore, sLOX-1 is suggested as a sensitive and specific bi-omarker in diagnosing acute coronary syndromes and acute aortic dissection. It has important implications for accelerating their identification and preventing potentially fatal complications in subjects with atherosclerosis-related conditions [98].
In another study, sLOX-1 levels were analyzed in overweight/obese children and ado-lescents, finding that these levels were elevated during and after puberty compared to the general population. Additionally, sLOX-1 was positively associated with inflam-matory markers and unfavorable cardiometabolic risk profiles, such as insulin re-sistance, dyslipidemia, and hypertension. These findings suggest that sLOX-1 could play an important role as an early biomarker of cardiometabolic risk and inflammation in overweight/obese children and adolescents [99].
On the other hand, patients with acute coronary syndromes (ACS) were examined to assess the role of vascular inflammation and biomarkers, specifically sLOX-1, in the development and complexity of coronary artery disease (CAD). Data were collected from patients admitted to the emergency department with unstable angina or NSTE-ACS. Elevated levels of high-sensitivity C-reactive protein (hs-CRP) and sLOX-1 were associated with more complex CAD, as per the modified Gensini score; this suggests that vascular inflammation, indicated by sLOX-1, could play a crucial role in risk pre-diction and ACS management, highlighting its significance as a novel predictive bi-omarker in individuals with this disease [100].
Incremental Value of sLOX-1 Over Established Biomarkers
- Early Detection: Cardiac troponins are the gold standard for diagnosing AMI due to their high sensitivity and specificity for myocardial injury. However, troponins may only elevate several hours after the onset of AMI. sLOX-1, by contrast, might rise earli-er in the course of endothelial activation and oxidative stress, potentially allowing for quicker diagnosis and intervention.
- Sensitivity to Subclinical Atherosclerosis: Unlike troponins and CK-MB, which increase in response to myocardial damage, sLOX-1 levels reflect endothelial dysfunc-tion and oxidative stress, which are early events in atherosclerosis; this makes sLOX-1 a potentially valuable biomarker for identifying subclinical atherosclerosis and pre-dicting cardiovascular events, even before structural heart damage occurs.
- Prognostic Value: Studies have suggested that elevated levels of sLOX-1 are asso-ciated with an increased risk of future cardiovascular events, providing prognostic in-formation beyond the acute setting; this could be particularly useful for risk stratifica-tion and long-term management of patients with coronary artery disease.
Integrating sLOX-1 into Clinical Practice
- Diagnostic Algorithms: Integrating sLOX-1 testing into routine diagnostic algo-rithms for AMI and stroke could complement the use of current biomarkers, potential-ly leading to earlier and more accurate risk stratification and therapeutic interven-tions. For example, an elevated sLOX-1 level in a patient with borderline troponin lev-els could prompt more aggressive management or further diagnostic testing.
- Feasibility: Integrating sLOX-1 testing is feasible only if rapid, cost-effective, and reliable assays are developed. Current enzyme-linked immunosorbent assay (ELISA) kits for sLOX-1 are primarily used in research settings and must be adapted for routine clinical use.
Challenges and Opportunities for Implementing LOX-1 as a Routine Biomarker
- Standardization: A significant challenge is the lack of standardized thresholds for sLOX-1 levels across different populations and clinical settings. Establishing these norms would require extensive validation studies.
- Cost and Accessibility: As with any new diagnostic tool, cost-effectiveness studies would be crucial to justify the inclusion of sLOX-1 testing in standardized diagnostic protocols, particularly in resource-limited settings.
- Education and Awareness: To ensure its effective use, clinicians must be educated about the implications of sLOX-1 levels and their integration into clinical decision-making processes.
- Regulatory Approval: Gaining regulatory approval for clinical use involves rig-orous testing and validation to meet safety and efficacy standards, which can be time-consuming and costly.
Future Prospects: The potential for sLOX-1 to become a routine biomarker shortly will largely depend on ongoing and future clinical trials that can definitively demonstrate its added value in clinical settings. With positive results, sLOX-1 could fill a crucial gap in the early diagnosis and risk stratification of cardiovascular diseases, ultimately leading to better patient outcomes.
In conclusion, it is essential to highlight the role of sLOX-1 measurements in clinical practice as they offer a window into a more precise evaluation of medical conditions related to the development and progression of diseases. Considering its potential, as found in the literature, it is highly feasible to integrate sLOX-1 testing into routine di-agnostic algorithms for acute myocardial infarction (AMI) and stroke. The introduction of sLOX-1 into these algorithms could present unique challenges and opportunities. On the one hand, we may face obstacles such as additional validation of its effectiveness in different clinical contexts and standardization of testing procedures. However, the successful integration of sLOX-1 could open new doors to earlier and more accurate detection of AMI and stroke, potentially improving clinical outcomes and reducing the burden on healthcare systems. Achieving this goal will require interdisciplinary collaboration, ongoing research, and careful assessment of associated benefits and risks. According to reported findings, significant advances in incorporating sLOX-1 as a routine biomarker could be seen shortly, marking a step forward in personalized and precision healthcare with simple tests such as ELISA in patient plasma.

Round 2
Reviewer 2 Report
Comments and Suggestions for Authors
All comments has been incorporated in the manuscript. No more comments
Thanks